# Transient inhibition of p53 enhances prime editing and cytosine base-editing efficiencies in human pluripotent stem cells

Mu Li[1,4], Aaron Zhong [1,4], Youjun Wu[1,4], Mega Sidharta[1,4], Michael Beaury[1], Xiaolan Zhao [2], Lorenz Studer [3] ✉ & Ting Zhou [1] ✉

Precise gene editing in human pluripotent stem cells (hPSCs) holds great promise for studying and potentially treating human diseases. Both prime editing and base editing avoid introducing double strand breaks, but low editing efficiencies make those techniques still an arduous process in hPSCs. Here we report that co-delivering of p53DD, a dominant negative fragment of p53, can greatly enhance prime editing and cytosine base editing efficiencies in generating precise mutations in hPSCs. We further apply PE3 in combination with p53DD to efficiently create multiple isogenic hPSC lines, including lines carrying GBA or LRRK2 mutations associated with Parkinson disease and a LMNA mutation linked to Hutchinson-Gilford progeria syndrome. We also correct GBA and LMNA mutations in the patient-specific iPSCs. Our data show that p53DD improves PE3 efficiency without compromising the genome-wide safety, making it feasible for safe and routine generation of isogenic hPSC lines for disease modeling.

Developments of the base editing (BE) and prime editing (PE) technologies for precise genome editing without resorting to double-stranded DNA breaks (DSBs) or donor DNA templates have greatly expanded the capabilities of gene editing technologies and mitigated safety concerns for applications in gene therapies[1]. Gene editing technologies in combination with human pluripotent stem cells (hPSCs) holds great potential for building human disease-in-a-dish models, drug discovery, and for cell-based gene therapy[2]. The CBE enzyme was created by the fusion of a dCas9 or Cas9 nickase and the cytidine deaminase enzyme (CBE). Working along with an sgRNA construct, CBE can mediate the direct conversion of C-to-T at a designated site in a genome[3–5]. The PE enzyme was engineered by the fusion of a Cas9 H840A nickase and an M-MLV reverse transcriptase. Working along with a pegRNA construct, PE can introduce all types of SNP changes, precise deletion, or insertions at a specific site of a genome[6]. While the application of CBE is currently still limited, in part by the restricted editing window and by the generation of bystander products, PE is considered to be highly versatile and was estimated to be able to correct ~89% of the known mutations that cause human diseases[6]. PE was also reported recently to be associated with a very low off-target effect (frequencies ~0.1 to 1.9%) in human cells[7]. Notwithstanding the powerful possibilities of CBE and PE, the generally low editing efficiencies of these tools in hPSCs make them still an arduous process to perform in a laboratory.

Previous studies reported a CRISPR/Cas9-induced, p53-mediated stress response and cell cycle arrest in hPSCs and other cell types[8,9]. Although no DSB is involved in PE or CBE theoretically, whether DNA binding or nicking at the editing site induced by PE or CBE will trigger a p53-dependent response is still unknown. Here, utilizing hPSCs, we asked the question of whether p53 is a general bottleneck in gene

[1]The SKI Stem Cell Research Facility, The Center for Stem Cell Biology and Developmental Biology Program, Sloan-Kettering Institute for Cancer Research, 1275 York Avenue, New York, NY 10065, USA. [2]Molecular Biology Program, Memorial Sloan Kettering Cancer Center, New York, NY 10065, USA. [3]The Center for Stem Cell Biology and Developmental Biology Program, Sloan-Kettering Institute for Cancer Research, 1275 York Avenue, New York, NY 10065, USA. [4]These authors contributed equally: Mu Li, Aaron Zhong, Youjun Wu, Mega Sidharta. ✉e-mail: studerl@mskcc.org; zhout@mskcc.org

editing, and if transient inhibition of p53 can also enhance the editing efficiency of PE or CBE.

## Results

### Transient inhibition of p53 by p53DD increases PE and CBE-mediated editing efficiencies in a vector reporter system

To find a tool that can inhibit the activity of p53 reliably and transiently, we adopted an episomal vector that expresses a dominant-negative version of p53. The p53DD vector (with a p53 carboxy-terminal dominant-negative fragment) was previously used in studying reprogramming and generation of integration-free human iPSCs[10,11]. We first examined the effect of p53DD in CRISPR/Cas9-mediated HDR using a POU5F1-GFP and SOX2-tdTomato hPSC knock-in system (Supplementary Fig. 1a, d). We found that addition of p53DD into the electroporation cocktail significantly increased HDR efficiency by a factor of two- to sixfold in H1 hESCs ($1.6 \pm 0.6\%$ versus $0.4 \pm 0.1\%$ for POU5F1-GFP knock-in; $3.2 \pm 0.2\%$ versus $1.5 \pm 0.2\%$ for SOX2-tdTomato knock-in) measured by flow cytometry (Supplementary Fig. 1a–f). Those results are consistent with previous findings that inhibiting p53 can increase the efficiency in CRISPR/Cas9-mediated HDR[8,9].

We then asked whether transiently inhibiting p53 can increase the editing efficiency mediated by PE or CBE in hPSCs. To this end, we first modified the original pCMV-PE2 enzyme vector[6] by replacing the CMV promoter with an EF1α promoter to avoid the potential silencing issue of CMV promoters in hPSCs[12]. We named the modified PE2 enzyme as pEF-PE2 and used it throughout this study. We next generated an exogenous reporter vector EF1α-GTG-GFP (GTG-GFP), which allows us to determine the precise editing efficiencies. We note that the vector contains a GFP cassette with a mutated start codon (GTG) under the control of an EF1α promoter. As a result, no GFP is expressed when the vector is electroporated into hPSCs alone. Upon conversion of "GTG" to "ATG" by correcting the mutated start codon, GFP expression will be turned on (Supplementary Fig. 2a). To perform this single nucleotide mutation in the system, pegRNA was constructed with the desired G-to-A mutation. Along with the PE enzyme (pEF-PE2), the duo works as PE2 tools. An additional nicking sgRNA vector was added to form the PE3 tools (Supplementary Fig. 2b). For CBE, we adopted two versions of reported CBE enzymes, including a highly efficient CBE enzyme BE3-FNLS[13], and another enzyme BE3-eA3A[14] with reduced RNA off-target and self-editing activities[15]. The sgRNA for CBE was designed to target the non-coding strand and introduce the position 5 "C-to-T" conversion, in order to create the GTG-ATG conversion on the coding strand of the "GTG-GFP" vector (Supplementary Fig. 2c).

To test whether p53DD can enhance the PE or CBE editing efficiencies, we co-electroporated the plasmids including the GTG-GFP reporter vector, and the editing tools (PE2, PE3, or CBE) to H1 hESCs with or without p53DD. By 24 h post-electroporation, we observed detectable levels of GFP expression across various PE and CBE conditions, while very little GFP background was seen in the cells electroporated with the GTG-GFP vector alone (Supplementary Fig. 2d). PE3 was notably more efficient than PE2 on turning on the GFP expression, but the addition of p53DD still increased the percentage of GFP+ cells in both PE2 and PE3 editing conditions by two- to threefold ($11.7 \pm 2.0\%$ versus $4.5 \pm 0.8\%$ for PE2; $18.4 \pm 2.1\%$ versus $5.0 \pm 0.6\%$ for PE3) (Supplementary Fig. 2d–f). Further, the CBE enzyme BE3-FNLS was more efficient than BE3-eA3A, and addition of p53DD also significantly increased the percentage of GFP+ in both BE3-FNLS and BE3-eA3A editing conditions by two- to threefold ($39.0 \pm 2.2\%$ versus $19.7 \pm 2.1\%$ for BE3-FNLS; $14.6 \pm 1.9\%$ versus $5.5 \pm 0.3\%$ for BE3-eA3A) (Supplementary Fig. 2e, g). To minimize potential variation in the efficiency of electroporation in the presence of p53DD, we incorporated a puromycin expression vector for co-electroporation in each condition. We then used puromycin to select for positively transfected cells at 24 h post-electroporation (Supplementary Fig. 3a). With this approach and at 48 h post-electroporation, flow cytometry analysis showed that the

addition of p53DD still consistently increased the percentage of GFP+% by two- to threefold among different PE2, PE3 or CBE conditions and reaching ~50% in BE3-FNLS + p53DD (Supplementary Fig. 3b–d). These results suggest that the p53DD effect was independent of potential electroporation variations.

### p53DD increases the PE and CBE- mediated genome editing efficiencies in an hESC reporter line

To measure the editing efficiency of PE or CBE in hPSCs at a specific genomic locus, we used a heterozygous H1-SOX2-P2A-H2B-tdTomoto (H1-H2B-tdTomato) reporter line which was created by knocking-in a tdTomato transgene fused with P2A and histone H2B (H2B) before the stop codon of SOX2 in H1 hESCs (Fig. 1a). The H1-H2B-tdTomato reporter line constitutively expresses tdTomato in hPSC culture medium (Fig.1a). PegRNA of PE was designed to insert a "TGA" stop codon in the H2B reading frame for PE tools (Supplementary Fig. 4a). CBE including a CBE enzyme BE3-FNLS and a sgRNA construct were designed to introduce a C-to-T mutation (at position 4) to create a stop codon (TAG) in H2B (Supplementary Fig. 4b). As a result, tdTomato expression will turn off when the PE or CBE editing is successfully performed (Fig. 1b, c).

Using this system, we electroporated PE2, PE3, or CBE with or without p53DD into the H1-H2B-tdTomato cells. At 48 h post-electroporation, a small portion of the cells became tdTomato-negative under the fluorescent microscope in different CBE conditions (Supplementary Fig. 4c). Flow cytometry analysis determined that the percentage of tdTomato-negative cells increased by approximately threefold with adding p53DD for CBE editing ($12 \pm 0.6\%$ versus $4.2 \pm 0.3\%$) (Fig. 1d, e). Intriguingly, more tdTomato negative cells were seen under a fluorescent microscope with the PE editing conditions (Fig. 1f), suggesting higher efficiency for inducing stop codon by using PE tools to insert TGA compared to C-to-T conversion conducted by CBE in this model system. Despite efficient baseline editing by PE2 and PE3 tools, p53DD was still able to increase the tdTomato turn-off rate (Fig. 1f). Flow cytometry analysis showed the percentage of tdTomato-negative cells doubled in the PE editing condition with p53DD ($26.1 \pm 0.35\%$ versus $12.3 \pm 0.36\%$ for PE2; $41.9 \pm 1.3\%$ versus $22.4 \pm 0.5\%$ for PE3) (Fig. 1g, h). Miseq (amplicon sequencing) was performed to identify the editing efficiency (Fig. 1i). We found that the addition of p53DD enhanced the on-target efficiency of C-to-T conversion by ~2.5-fold of that induced by CBE tool alone ($12.6 \pm 0.13\%$ versus $5.3 \pm 0.2\%$) (Fig. 1j). For PE, the on-target efficiency of TGA insertion was enhanced by two- to threefold in the presence of p53DD comparing to PE2 or PE3 tools alone ($36.2 \pm 0.68\%$ versus $13.3 \pm 0.1\%$ for PE2; $47.3 \pm 1.3\%$ versus $23.7 \pm 0.3\%$ for PE3) (Fig. 1k). Therefore, the on-target editing efficiencies analyzed by Miseq were consistent with flow cytometry analysis using H1-H2B-tdTomato reporter cells. Together, these results suggest that p53DD can enhance the on-target editing efficiency mediated by both PE and CBE. We also looked at the potential indels at sgRNA or pegRNA nicking sites from the Miseq analysis data. We found that p53DD also increased the indel frequency at the sgRNA nicking site of CBE editing ($1.6 \pm 0.1\%$ versus $0.6 \pm 0.1\%$) (Fig. 1j). On the other hand, almost no indels were detected at pegRNA nicking sites with PE2 or PE3 with or without p53DD (indel frequency <0.2%) (Fig. 1k), suggesting the low byproduct potential of PE.

CBE has been shown to cause deamination of cytosines in DNA and RNA independent of sequence-specified CBE binding[16–18]. To investigate the role of p53 inhibition on potential off-target effects of CBE, we performed transcriptome-wide and genome-wide sequencing for CBE-edited cells with or without p53DD. After creating a stop codon in the H2B reading frame by CBE with or without p53DD in the H1-H2B-tdTomato reporter cells, we isolated the edited cell population (tdTomato negative) by FACS sorting (Supplementary Fig. 5a). RNA and genomic DNA were extracted from these cells for RNA-seq and whole-genome sequencing (WGS),

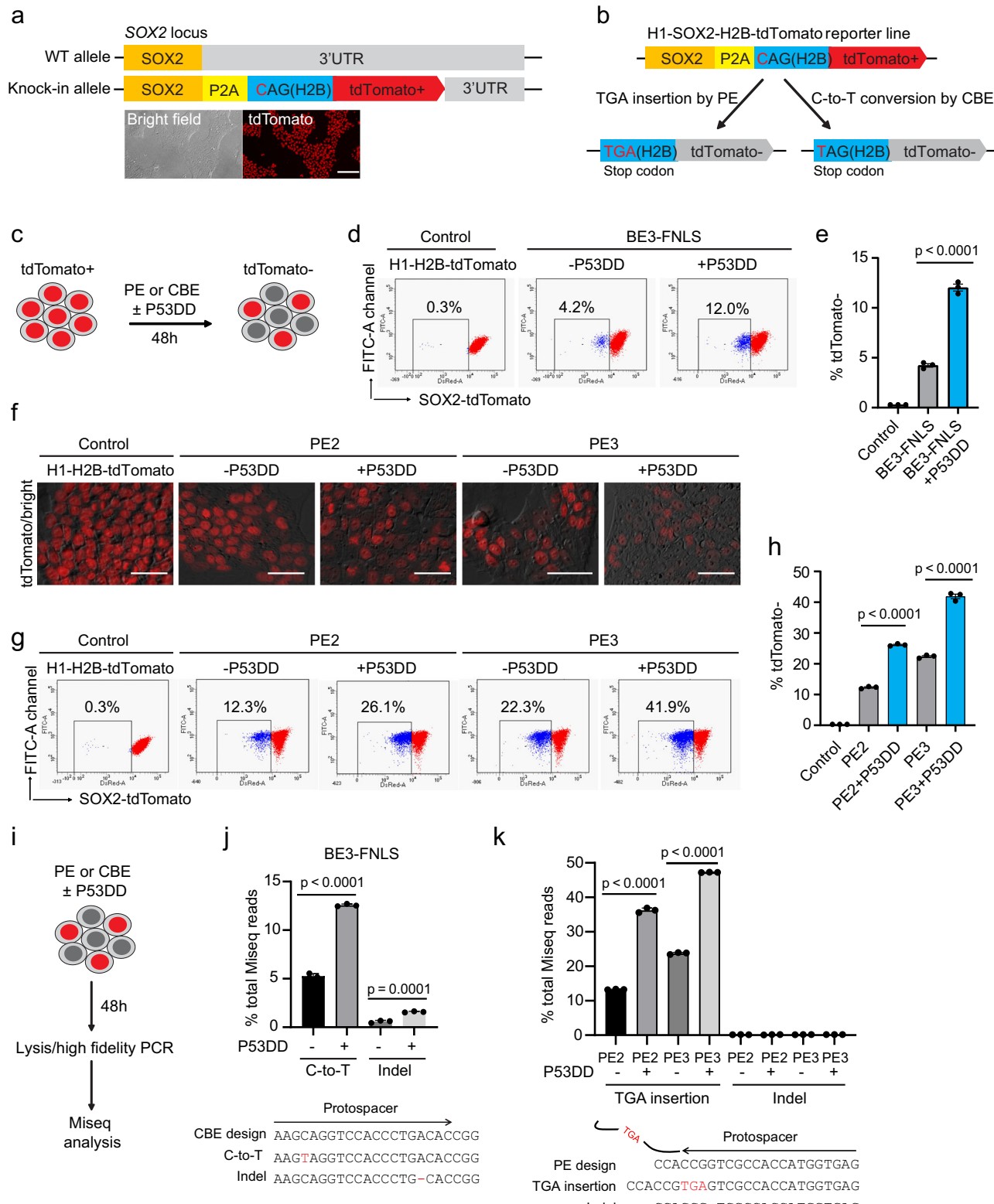

respectively. The RNA-seq analysis demonstrated that CBE with or without p53DD showed similar C-to-U edits (Supplementary Fig. 5b), indicating that p53DD did not increase cytosine deamination-induced off-target effects in RNA. Similarly, WGS analysis showed that total single-nucleotide variant (SNVs) as well as C:G to T:A SNVs are comparable with or without p53DD, though under both conditions (with and without -p53DD), we observed a slight increase (Supplementary Fig. 5c).

Electroporation is the most widely adopted delivery method for conducting genome editing in hPSCs[19,20]. Here, we test additional delivery methods by using Lipofectamine Stem Transfection Reagent (LipoStem)[21], to co-transfect PE2, PE3, or CBE with or without p53DD into H1-H2B-tdTomato endogenous reporter cells (Supplementary Fig. 6a). Despite the lower editing efficiency with LipoStem compared to electroporation, p53DD still increased ~2× of PE or CBE editing efficiency, consistent with the results obtained with electroporation

**Fig. 1 | p53DD increased the PE and CBE editing efficiencies in an H1-H2B-tdTomato reporter line. a** A scheme of the H1-H2B-tdTomoto reporter line, which contains a heterozygous P2A-H2B-tdTomato transgene knocked in to the SOX2 locus in H1 hESCs. The bright field and fluorescent images indicate the reporter line constitutively expresses tdTomato. Three fluorescent microscopic images were evaluated from 3 different cell passages. Scale bars, 250 μm. **b** Schematics of an "H2B-tdTomato" endogenous reporter assay for evaluating PE and CBE editing efficiency. **c**. Schematics of evaluating the effects of adding the p53DD plasmid into the PE and CBE editing tool kits respectively, based on "H2B-tdTomato" endogenous reporter assay. tdTomato turn-off rate was as the read out. **d, e** Flow cytometry analysis (**d**), and the quantification (**e**) of tdTomato-negative population % at 48 h post-electroporation with CBE tools, with or without p53DD. Unedited H1-H2B-tdTomato cell line was used as a control *n* = 3 independent electroporation reactions for each condition. Values presented as mean ± S.D. *p* values were calculated by ordinary one-way ANOVA. **f–h**, Fluorescent microscopic analysis

(**f**), flow cytometry analysis (**g**) and the quantification (**h**) of the tdTomato-negative population% at 48 h post-electroporation with PE2 or PE3 tools, as well as with or without p53DD. Scale bars, 100 μm. Unedited H1-H2B-tdTomato cell line was used as a control. *n* = 3 independent electroporation reactions for each condition. Values presented as mean ± S.D. *p* values were calculated by ordinary one-way ANOVA. **i** Schematics of Miseq analysis of edited H1-H2B-tdTomato cells by PE or CBE with or without p53DD. **j** Miseq analysis of C-to-T conversion and indels created at the sgRNA nicking site induced by CBE tools, with or without p53DD. *n* = 3 independent electroporation reactions for each condition. Values presented as mean ± S.D. *p* values were calculated by ordinary one-way ANOVA. **k** Miseq analysis of TGA insertion and indels around the pegRNA nicking site induced by PE tools, with or without p53DD. *n* = 3 independent electroporation reactions for each condition. Values presented as mean ± S.D. *p* values were calculated by ordinary one-way ANOVA. The source data of **e, h, j, k** are provided in Source Data file.

(Supplementary fig. 6b–f). These results suggest that p53DD can enhance PE and CBE editing efficiency and the effect is independent on electroporation.

In the study, we adopted a p53DD episomal vector (pCE-p53DD) that co-expresses a multifunctional viral protein, EBNA-1. To address whether the effects of the p53DD construct are derived from the p53DD fragment or the EBNA-1 fragment, we modified the original pCE-p53DD construct to generate a new EBNA-1 expression construct (EBNA-1), and a new p53DD-only expression construct (p53DD-only) (Supplementary Fig. 7a). Importantly, both new constructs and the original pCE-p53DD are all under the control of an identical CAG-promoter (Supplementary Fig. 7a). We then tested the EBNA-1 and p53DD-only constructs in the PE and CBE co-electroporation experiments, using the original pCE-p53DD as control. We found co-electroporation of p53DD-only, but not the EBNA-1 expression construct increased the PE and CBE editing efficiency, with highly comparable results to the pCE-p53DD control (Supplementary Fig. 7b–e). These data demonstrate that EBNA-1 alone cannot increase the PE or CBE editing efficiency. Either p53DD episomal vector or a simple "p53DD-only" expression vector can enhance the PE or CBE editing efficiency to a similar level.

Furthermore, we also evaluated the inhibition of p53 by three alternative mechanisms in PE and CBE editing. First, we tested p53 siRNA, and found co-electroporation of p53 siRNA did not increase PE or CBE editing efficiency (Supplementary fig. 8a–d), which is likely due to the different kinetics between genome editing and RNA silencing[22], and that p53 is mainly regulated at the protein level[23,24]. Second, we tested p-nitro-Pifithrin-α (a cell-permeable PFT-α), a widely used small-molecule p53 inhibitor[25], and we found that only a high dose of p-nitro PFT-α (10 μM) can increase the CBE, but not PE2 or PE3 editing efficiency (Supplementary Fig. 8a, e–g). A recent study discovered that PFT-α displays other p53-independent activity in cells and fails to prevent the p53 effects on the cell cycle and apoptosis in some cases. The report further suggests caution in using PFT-α to study p53-dependent processes[26]. Our data show that either p53 siRNA or PFT-α cannot replace p53DD to enhance the PE or CBE editing efficiency in hPSCs. Lastly, we tested co-electroporation of a MDM2 expressing vector[27] with PE or CBE editing tools. MDM2 is the main E3 ligase for p53 ubiquitination in cells, which mediates poly-ubiquitination of p53 and directs it for proteasomal degradation[24,28,29]. We found co-electroporation of MDM2 can mimic the effect of p53DD in hPSCs by significantly improving the PE or CBE editing efficiency (Supplementary Fig. 8h–l).

## p53DD increased PE efficiency for precise insertion and deletion at endogenous loci

We further tested whether p53DD can enhance the editing efficiency in previously reported genome loci editable by PE. We tested a pegRNA targeting human HEK3 locus for "CTT" insertion as reported

previously[6]. We also designed a pegRNA for insertion of a "LoxP" (34 bp), or for deletion of "GT" (2 bp) in H1 hESCs (Fig. 2a). Miseq analysis showed that p53DD increased the efficiency of CTT insertion by two- to threefold in either PE2 or PE3 editing (9.2 ± 0.8% versus 3.5 ± 0.8% for PE2; 23.2 ± 0.7% versus 9.5 ± 3.0% for PE3) (Fig. 2b). Consistently, p53DD also significantly enhanced the efficiency of "LoxP" insertion in either PE2 or PE3 editing (10.1 ± 0.1% versus 5.1 ± 0.6% for PE2; 12.5 ± 1.8% versus 3.8 ± 0.2% for PE3) (Fig. 2c). For "GT" deletion, Miseq analysis showed that p53DD slightly increased the efficiency for PE2 and significantly increased editing efficiency (>2fold) for PE3 (1.64 ± 0.23% versus 1.18 ± 0.14% for PE2; 15.94 ± 0.71% versus 6.95 ± 1.79% for PE3) (Fig. 2d).

Low levels of indels frequency (all <1% in total sequencing reads) were detected at the nicking site of the pegRNA (indel 1) or at the nicking site of the additional nicking sgRNA of PE3 tools (indel 2) among all the PE editing conditions with or without p53DD at HEK3 locus (Fig. 2b–d). To check more endogenous sites for potential indels induced by p53DD for "CTT" insertion, we extended the quantification window to 10 bp size which calculated 20 sites adjacent to the pegRNA nicking site (Supplementary Fig. 9a). We calculated non-edited frequency, total indel frequency, desired perfect "CTT" insertion, and imperfect "CTT" insertion ("CTT" inserted but carried other mutations) in these 20 sites using CRISPResso2 (Supplementary Fig. 9b). The data showed that PE3 increased both desired and undesired editing frequencies compared to PE2, which is consistent with the previous report[6]. The presence of p53DD significantly improved the desired editing efficiency in both PE2 and PE3; the presence of p53DD also increased the total indel and the imperfect "CTT" insertion frequencies, albeit frequencies still represented only a small percentage overall (<1% of each).

We then evaluated the indels frequencies at the loci with guide-target mismatch ≤3, compared to the pegRNA or nicking sgRNA targets of PE3 (mismatch sgRNA sequences are listed in Supplemental Table 5a). Miseq analysis showed no or very low levels of indels were detected at those endogenous loci, in PE3 with or without p53DD editing conditions (Supplemental Table 5b). These results suggest co-electroporation with p53DD did not increase the indel frequency at those potential off-target loci in PE3 editing for "CTT" insertion.

Furthermore, we evaluated a disease-relevant deletion, the SERPING1 (c.351delC) mutation, linked to the development of hereditary angioedema in patients[30]. PE2 or PE3 tools with or without p53DD were electroporated to H1 hESCs to introduce SERPING1 (c.351delC). Miseq analysis showed that p53DD significantly increased the precise "C" deletion in either PE2 or PE3 editing (2.72 ± 0.53% versus 1.12 ± 0.64% for PE2; 24.33 ± 1.05% versus 12.26 ± 0.60% for PE3) (Fig. 2e). Meanwhile, low frequency of indel 1 and indel 2 (<0.2%) were detected (Fig. 2e), further suggests the feasibility of including p53DD in generating disease relevant deletions in hPSCs.

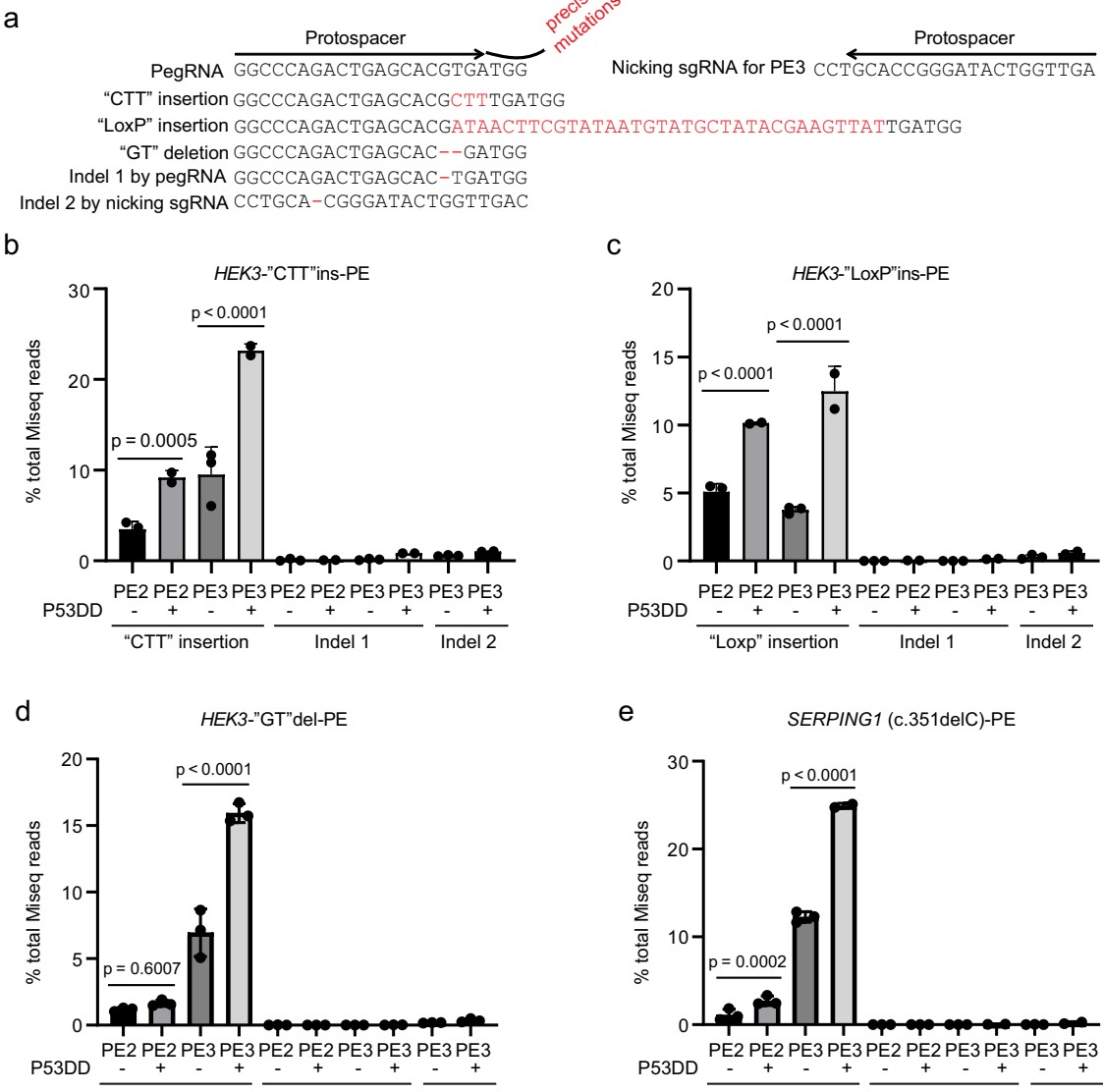

**Fig. 2 | p53DD increased PE efficiency for precise insertion and deletion at endogenous loci. a** Schematics of pegRNA design and PE tools for precise "CTT" or "Loxp" insertion or "GT" deletion at HEK3 locus in H1 hESCs. The PegRNA was designed to include a protospacer sequence that targets the HEK3 locus, a RT template of the 3′ extension was designed to include the edit: CTT (3 bp) or LoxP (34 bp) insertion, or "GT" (2 bp) deletion. Additional nicking sgRNA targets were designed for PE3. Listed sequences indicate the "CTT" or "Loxp" insertion or "GT" deletion (on-target products), potential indels at the pegRNA nicking site (indel 1), and potential indels at the nicking sgRNA targeting site (indel 2). On target, Indel 1 and Indel 2 frequencies were analyzed in **b**–**d**. **b**–**d** Miseq analysis of editing efficiency at the HEK3 locus for "CTT" insertion (**b**), "LoxP" insertion (**c**), and "GT"

deletion (**d**) in H1 hESCs using PE tools with/without p53DD, as well as the potential indels frequency. For insertion: $n = 3$ independent electroporation reactions for PE2 and PE3, without p53DD conditions. $n = 2$ independent electroporation reactions for PE2 and PE3, with p53DD conditions. For deletion: $n = 3$ independent electroporation reactions for each condition. Values presented as mean ± S.D. $p$ values were calculated by ordinary one-way ANOVA. **e.** Miseq analysis of editing efficiency for introducing SERPING1 (c.351delC) in H1 hESCs using PE tools with/without p53DD. $n = 2$ independent electroporation reactions for PE3 with p53DD conditions. $n = 3$ independent electroporation reactions for all other conditions. Values presented as mean ± S.D. $p$ values were calculated by ordinary one-way ANOVA. The source data of **b**–**e** are provided in Source Data file.

## p53DD boosts PE efficiency for the generation of hPSC isogenic models for PD and HGPS

Isogenic hPSC lines with disease-relevant mutations are powerful tools for studying the genetic mechanisms of diseases. Given the great potential of PE to perform versatile and precise editing in hPSCs, we further evaluated applying PE to generate isogenic hPSC lines for Parkinson's disease (PD) and Hutchinson-Gilford progeria syndrome (HGPS). GBA (c. 1226 A > G, p. N370S) and LRRK2 (c. 6055 G > A, p. G2019S) are two common familial mutations in PD[31], and LMNA (c.1824 C > T; p.G608G) mutation is known to cause HGPS[32]. We tried to introduce these three mutations, respectively, to an iPSC line

generated from a healthy individual (MSK-SRF001-iPSCs) (Fig. 3a). Towards this end, we used PE2 and PE3 with or without p53DD to conduct the gene editing. PE2 or PE3 with or without p53DD was electroporated to MSK-SRF001-iPSCs. At 48 h after electroporation, we collected the cells for Miseq analysis and measured the editing efficiency. Miseq analysis showed that PE2 alone had a very low efficiency of inducing these mutations (all < 1%). The addition of p53DD was not able to increase the editing efficiency significantly (Fig. 3b–d), which was not consistent with the findings in the H1-H2B-tdTomato reporter system, as well as for the precise insertion and deletion studies. This may suggest PE2-editing efficiencies could vary depending

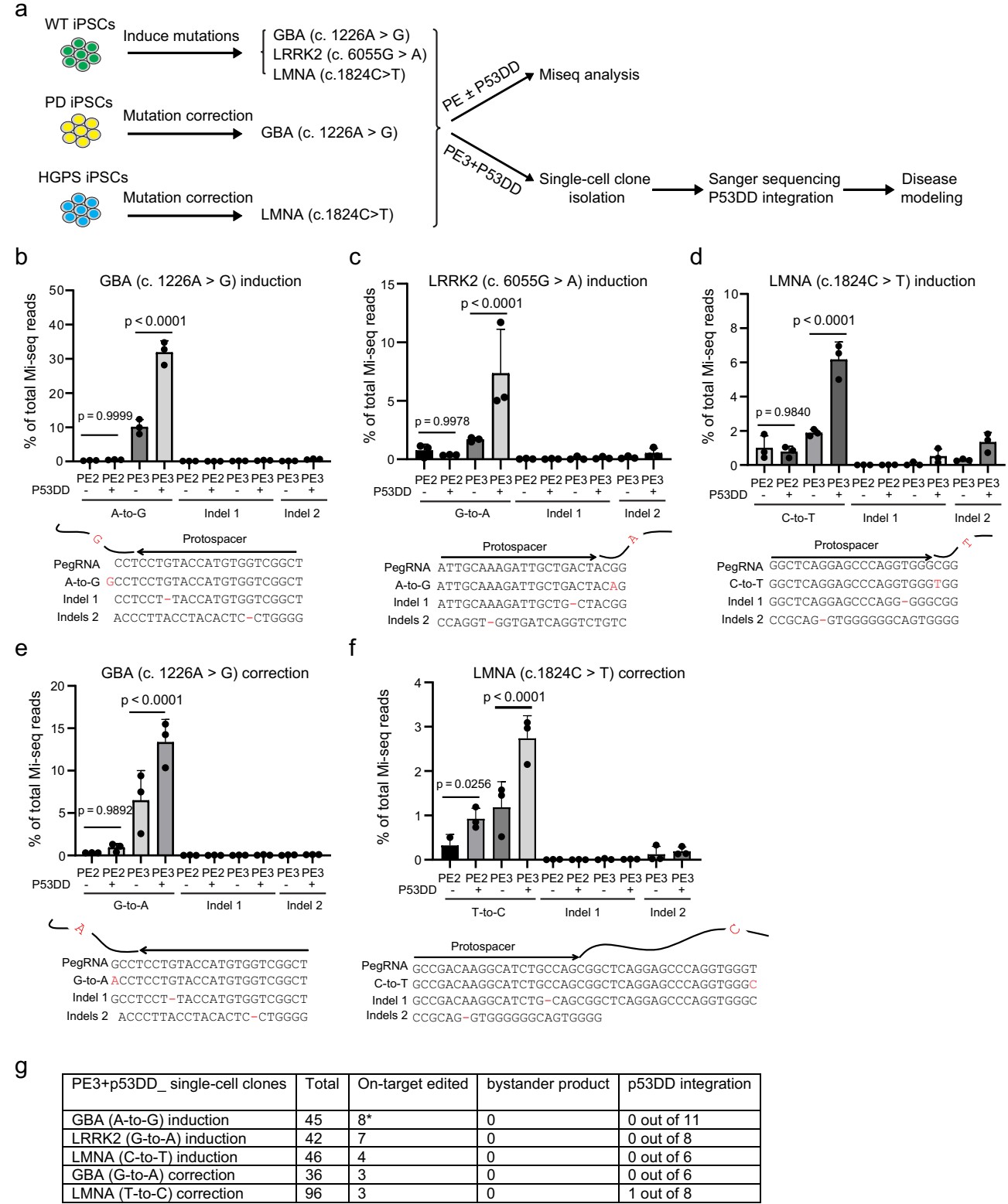

g

| PE3+p53DD_ single-cell clones | Total | On-target edited | bystander product | p53DD integration |
|---|---|---|---|---|
| GBA (A-to-G) induction | 45 | 8* | 0 | 0 out of 11 |
| LRRK2 (G-to-A) induction | 42 | 7 | 0 | 0 out of 8 |
| LMNA (C-to-T) induction | 46 | 4 | 0 | 0 out of 6 |
| GBA (G-to-A) correction | 36 | 3 | 0 | 0 out of 6 |
| LMNA (T-to-C) correction | 96 | 3 | 0 | 1 out of 8 |

*4 heterozygous and 4 homozygous clones.

on the loci. We suspect that the chromatin environment at these loci may disfavor editing of PE2. The status and the role of RNA polymerase in gene editing would also be interesting to examine in future studies.

On the other hand, the PE3 editing efficiency was notably higher than PE2, but variant among the three mutations ($10.1 \pm 2.2\%$ for GBA, $1.7 \pm 0.2\%$ for LRRK2, $1.9 \pm 0.2\%$ for LMNA) (Fig. 3b–d). The addition of p53DD could enhance the efficiency of PE3 by three- to fourfold for the

three mutations ($31.9 \pm 3.4\%$ for GBA, $7.3 \pm 3.8\%$ for LRRK2, $6.2 \pm 1.0\%$ for LMNA) (Fig. 3b–d). Very few indels were detected at the pegRNA nicking site (indel 1) or the additional sgRNA nicking site (indel 2) for inducing GBA and LRRK2 mutations regardless of p53DD (Fig. 3b, c). We observed the p53DD moderately increased the indel 1 and indel 2 frequency for inducing LMNA mutation, though still resulting in relatively low level (indel 1 frequency $0.5 \pm 0.4\%$, and indel 2 frequencies

**Fig. 3 | p53DD increased PE editing efficiency for generation of diseases-specific isogenic hiPSC lines. a** Schematics of applying PE tools to create isogenic hiPSC line. Specifically, PE2 or PE3 with or without p53DD was used to introduce PD mutations: GBA (c. 1226 A > G, p. N370S) and LRRK2 (c. 6055 G > A, p. G2019S), as well as HGPS mutations: LMNA (c.1824 C > T; p.G608G) in an WT iPSC line (MSK-SRF001-iPSCs) respectively. PE2 or PE3 with or without p53DD were also applied to correct the heterozygous GBA (c. 1226 A > G, p. N370S) mutation in a PD patients' iPSC line (756-iPSCs), and correct the heterozygous LMNA (c.1824 C > T; p.G608G) mutation in an HGPS patients' iPSC line (972-iPSCs). Miseq analysis was used to evaluate the gene editing efficiency in each condition. Single-cell clones with desired edits and the isogenic controls were isolated from cells edited with PE3 + p53DD for further characterization. **b–f** Miseq analysis of on-target efficiencies and indel frequencies using PE editing tools with/without p53DD, including GBA (c.1226 A-to-G) (**b**), LRRK2 (c. 6055 G-to-A) (**c**), LMNA (c.1824 C-to-T) (**d**) induction in MSK-SRF001 hiPSCs, as well as the GBA (c. 1226 G-to-A) mutation correction in 756-iPSCs (**e**), and the LMNA (c.1824 T-to-C) mutation correction in 972-iPSCs (**f**). Sequences listed below each graph indicate the on-target products, the indels at the pegRNA nicking site (indel 1), and the indels at the nicking sgRNA targeting site (indel 2) shown by Miseq results. *n* = 3 independent electroporation reactions for each condition. Values presented as mean ± S.D. ns indicates a non-significant difference. *p* values were calculated by ordinary one-way ANOVA. **g** A summary table of diseases' single-cell clone screening by sanger sequencing. Three unmodified (isogenic control) clones and indicated a number of on-target edited clones were isolated and expanded. The number of the clones that carried bystander product or detected p53DD plasmid integration was also shown. The source data of **b–f** are provided in Source Data file.

1.3 ± 0.6%) (Fig. 3d). Further, we isolated and expanded 3 single-cell clones with LMNA (c.1824 C > T) mutations and three isogenic control clones from the PE3 + p53DD editing condition. No indels (indel 1 or indel 2) or other bystander products were seen in these six single-cell clones, as confirmed by Sanger sequencing (Supplementary Fig. 10a), suggesting the low levels of indels detected by Miseq were not captured in the selected single-cell clones.

We also isolated and expanded eight single-cell clones carrying the GBA (c. 1226 A > G) mutation, 4 single-cell clones carrying the LRRK2 (c. 6055 G > A) mutations, as well as isogenic WT control clones for each mutation. No indels (indel 1 or indel 2) or other bystander products were detected in these clones (Supplementary Fig. 11a, b). Interestingly, we obtained 4 homozygous and 4 heterozygous GBA (c. 1226 A > G) clones out of a total of 45 with PE3 + p53DD, and only obtained clones with heterozygous mutations for LRRK2 (7 out of 42) or LMNA (4 out of 46) (Fig. 3g and Supplementary table 1). This may suggest PE3 + p53DD was able to create heterozygous and homozygous mutations depending on the editing efficiency. Together, our results suggest p53DD can significantly increase PE3 editing efficiencies without inducing indels or bystander products in the edited single-cell clones.

Finally, we tested using PE and p53DD for gene correction in patients' iPSCs. PegRNAs were designed to target a GBA (c. 1226 A > G) mutation in a PD patient-specific iPSC line (756-iPSCs), and a LMNA (c.1824 C > T) mutation in a HGPS patient-specific iPSC line (972-iPSCs), respectively (Fig. 3a). Miseq analysis showed that p53DD increased the PE2 editing efficiency slightly in both cases, but significantly enhanced the PE3 editing (13.4 ± 2.7% versus 6.5 ± 3.5% for GBA correction; 2.7 ± 0.5% versus 1.2 ± 0.6% for LMNA correction) (Fig. 3e, f). Again, very few indels (frequency < 0.2% for indel 1 or indel 2) were observed in both gene corrections with or without p53DD (Fig. 3e, f). We picked and expanded 3 GBA mutation corrected and 4 LMNA mutation-corrected clones, as well as their isogenic non-edited single-cell clones (Supplementary Fig. 11c, d). Consistently with previous obversions, no indels (indel 1 or indel 2) or other bystander products were detected in these single-cell clones (Fig. 3g).

To comprehensively assess the genome-wide off-target effect of PE3 with or without p53DD, we performed whole-genome sequencing (WGS) to identify genome-wide SNVs and indels in two GBA mutation corrected single-cell clones generated by PE3 editing (clone #36 and #66), and in PE3 with P53DD editing (clone #1–5 and #27). The parental patient iPSCs (756-iPSCs) were used as the background control for WGS (Supplementary Fig. 12a). WGS analysis showed that the total number of new base substitutions or indels was not increased by PE3 edited, or PE3 with p53DD-edited single-cell clones, compared to parental unedited control (Supplementary Fig. 12b, c). These data indicated that p53DD improved PE efficiency without compromising the genome-wide safety of PE.

The episomal expression system contains OriP/EBNA1 components of Epstein–Barr virus, which enables relatively high and long-term gene expression in the host cells, while still allowing for a gradual loss of vector from proliferating cells[33]. Transfection or electroporation of plasmids may be associated with a low level of genome integration. We asked whether the p53DD plasmid could integrate into the genome of edited cells. For this purpose, we used genomic DNA from the edited clones and PCR to determine the presence or absence of p53DD sequence in the clones edited with PE3 + p53DD conditions. We analyzed a total of 39 clones. PCR analysis demonstrated that the p53DD vector was absent in the genomic DNA in 38 out of 39 clones (Supplementary Fig. 13a–g), while we found one clone (972-iPSC-LMNA correction #87) with detectable integration of p53DD vector in the genomic DNA (Supplementary Fig. 13g). While the integration frequency is low (1 out of 39), this suggests that an additional characterization step for p53DD integration may be required for the clones edited with the p53DD plasmid.

## Discussion

PE performs versatile editing (in our case, we performed point mutations, 3 bp and 34 bp insertions, 1 bp and 2 bp deletions), which is a powerful tool for precise editing in hPSCs. However, when on-target efficiency is low in some cases, it is time-consuming and costly to pick and identify the correctly edited hPSC single-cell clones. For example, in our initial tests for LMNA (c.1824 C > T) correction in 972-iPSCs, we did not obtain any single-cell clone with mutation corrected, within ~200 single-cell clones isolated using PE3 alone. With the addition of p53DD, we obtained three corrected single-cell clones out of a total 96, which makes this approach feasible.

Here, we describe an optimized method for conducting gene editing with PE and CBE in hPSCs by transient inhibition of p53. We found the p53DD plasmid can be conveniently applied to established PE or CBE protocols for efficient gene editing in hPSC, including SNP mutation, insertion, and deletion. As proof of concept, we applied PE3 + p53DD approach to generate multiple disease-relevant isogenic hPSCs lines for the study of PD and HGPS.

Our initial testing suggested that the ABE editing tools are generally inefficient in hPSCs, making them less desirable for the routine generation of disease-relevant isogenic hPSCs. However, it would be interesting to test if p53DD can enhance ABE editing efficiency in other cell types such as primary cells or cancer cells.

Unlike CRISPR/Cas9, PE or BE should not lead to DSB. However, using qRT-PCR, we found that mRNA levels of CDKN1A (Cyclin-dependent kinase inhibitor 1A, encoding p21) were significantly increased in CBE or PE3-edited cells, and less so in PE2-edited cells. The addition of p53DD largely inhibited this effect (Supplementary Fig. 14). We performed PE and CBE editing experiments with a CDK4/6 inhibitor (Palbociclib)[34], or a DNA synthesis inhibitor (Thymidine) that can arrest cells at G1/S phase transition[35]. We found Palbociclib or Thymidine decreased PE and CBE editing efficiency in hPSCs (Supplementary Fig. 15a–e), suggesting that cell cycle arrest can decrease the PE or CBE editing activity.

Interestingly, a previous study showed that CRISPR/Cas9 induces cell cycle arrest driven partly by Cas9 binding to DNA[36]. Since PE and

CBE enzymes contained the Cas9 nickase that binds to DNA, such binding could cause DNA stress and thereby activate p53 and P21. Here, we generated dCas9-based CBE vector (eA3A-BE2) (Supplementary Fig. 16a). We found electroporation of eA3A-BE2 editing tools did not induce p21 transcription (Supplementary Fig. 16b). Meanwhile, co-electroporation with p53DD did not increase eA3A-BE2 editing efficiency (Supplementary Fig. 16c, d). These results indicated that DNA nicking, but not DNA binding per se, by Cas9 causes the induction of p53-p21 axis. It is worthwhile to note that given that BE2 editing efficiency was low in hPSCs, further testing using other editing systems may help to better understand the mechanism by which p53 inhibition can affect gene editing efficiency in hPSCs.

In summary, our data support a model whereby co-electroporation of p53DD removes the roadblock of cell cycle inhibition induced by PE or CBE editing tools and increases the editing efficiency of PE or CBE in hPSCs (Supplementary Fig. 17); however, we do not exclude a possible effect of p53DD on increasing overall cell fitness for exposure to these gene editing tools.

MMR inhibition via co-delivering MLH1dn was recently shown to improve PE efficiencies by Chen et al.[37]. A future research direction is to examine whether combining this approach with p53 inhibition could further enhance PE efficiency. It will be also interesting to evaluate whether co-deliver of p53DD facilitates gene editing in primary cells or in vivo.

Further studies dissecting the mechanistic role of p53, cell cycle, and DNA damage during the process of PE or CBE-mediated gene editing may provide additional insight into this bioengineering approach, and fully unlock its therapeutic potential.

## Methods

This research complies with all relevant ethical regulations. Informed consent was obtained and the study was approved by our Institutional Review Board from Memorial Sloan-Kettering Cancer for MSK-SRF001-iPSCs generated from urine cells isolated from a healthy donor. In addition, the use of all the hESCs and hiPSCs from this study was approved by the Tri-institutional (MSKCC, Weill-Cornell, Rockefeller University) Stem Cell Research Oversight (SCRO) Committee.

### hPSC lines and cell culture

H1 hESCs were purchased from WiCell Institute. The H1 SOX2-P2A-H2B-tdTomato reporter line was generated by knocking in a P2A-H2B-tdTomato transgene before the stop codon at the SOX2 gene locus using CRISPR/Cas9-based HDR[38]. The MSK-SRF001-iPSCs were generated from urine cells of an apparently healthy donor using a previously reported method[39]. The 972-iPSCs were previously generated from an HGPS patient's fibroblast line purchased from Coriell Institute (AG01972)[40], and the 756-iPSCs were generated from a PD patient's fibroblast line purchased from Coriell Institute (ND29756). All the hPSCs have been fully characterized and are routinely cultured on Matrigel (Fisher Scientific 08-774-552) in Stemflex Medium (Thermo Fisher A3349401). Cells were maintained at 37 °C with 5% $CO_2$. For regular passaging, cells were detached and passaged with 0.5 mM EDTA (Fisher Scientific MT-46034CI) at room temperature for 5 min.

### Plasmids construction

pCE-p53DD (p53DD) was a gift from Shinya Yamanaka (Addgene plasmid # 41856)[10]. The pCMV-PE2 (Addgene plasmid # 132775) and pU6-pegRNA-GG-acceptor (Addgene plasmid #132777) plasmids were a gift from David Liu[6]. The modified PE2 enzyme vector pEF-PE2 was generated by switching the CMV promoter of pCMV-PE2 to EF1α promoter by Gibson cloning. PegRNAs were cloned into the pU6-pegRNA-GG-acceptor vector by Golden Gate assembly following the protocol previously published[6]. The high efficient BE3-FNLS vector was a gift from Lukas Dow (Addgene plasmid # 112671)[13], and the reduced RNA off-target eA3A-BE3 vector a gift from Keith Joung (Addgene plasmid #

131315)[15]. The pX330-U6-Chimeric_BB-CBh-hSpCas9 (Addgene plasmid #42230) was a gift from Feng Zhang[41]. The pSPgRNA (Addgene plasmid # 47108) was a gift from Charles Gersbach[42]. The sgRNA target sequences were cloned into the pX330-U6-Chimeric_BB-CBh-hSpCas9 vector for POU5F1-GFP or SOX2-tdTomoto knock-in. The nicking sgRNA target sequences were cloned into the pSPgRNA vector for CBE or PE3-related experiments (we named the pSPgRNA-based sgRNA constructs as "Lsg- target" in this manuscript). The pEF-GFP was a gift from Connie Cepko (Addgene plasmid # 11154)[43]. pcDNA3 MDM2 WT was a gift from Mien-Chie Hung (Addgene Plasmid #16233)[27]. A "GTG-GFP" exogenous reporter vector was created by mutating "A" to "G" in the start codon of the pEF-GFP vector, and also removing extra "C" in the editing window of CBE. PUC57-puro was generated by cloning the PGK-Puro cassette into the vector pMD-18T (TaKaRa).

### sgRNA and pegRNA design

We used the Benchling CRISPR Design Tool[44] (https://www.benchling.com/#) to design all the sgRNA targets used in this study. The sgRNA targets designed for CRISPR/Cas9-based HDR or CBE or PE3-related experiments are listed in Supplementary Table 2. For pegRNA design, we first used the Benchling CRISPR Design Tool to find the best spacer sequence near the editing locus. Then, the PBS and RT templates of pegRNAs were designed using the criteria listed in previously published[6]: PBS = -13 nt, RT template = -10 nt, and the first base of the 3′ extensions is not C. For PE3, another sgRNA was designed for DNA nicking at the complementary strand at ~ 60-100 bp downstream from the pegRNA cutting site. The spacer and 3′ extension sequences of pegRNAs are listed in Supplementary Table 3.

### hPSC Electroporation

hPSCs were dissociated using Accutase (Innovative Cell Tech. AT104) at 37 °C for 10 min, followed by the addition of Stemflex Medium, and centrifuged at 120 × g for 3 min. We used 250,000 cells for a small electroporation reaction (Lonza V4XP-3032) or 1 × 10⁶ cells for a large reaction (Lonza V4XP-3034), following the manufacturer's instructions. The reactions were performed using Lonza 4D-Nucleofector X Unit, with the setting Pulse Code "CB-150". After electroporation, the cells were seeded into one well of a 48-well plate for a small reaction, or one-well of a six-well plate for a large reaction, if not otherwise specifically indicated.

### An "GTG-GFP" exogenous reporter vector for evaluating CBE and PE efficiency

The sgRNA and pegRNA constructs were designed to convert "GTG" to "ATG" start codon of GTG-GFP vector. To compare the PE and CBE editing efficiency with or without p53DD in this exogenous reporter assay, nine different combinations of plasmids were electroporated into H1 hESCs using the small electroporation reaction described above. 1 μg of each plasmid was used.

Table of plasmids used to compare the PE editing efficiency using exogenous reporter vector.

| Condition | Plasmids combination | | | |
|---|---|---|---|---|
| Control | GTG-GFP | | | |
| BE3-FNLS | GTG-GFP | BE3-FNLS | Lsg-GTG-GFP | |
| BE3-FNLS + p53DD | GTG-GFP | BE3-FNLS | Lsg-GTG-GFP | p53DD |
| BE3-eA3A | GTG-GFP | BE3-eA3A | Lsg-GTG-GFP | |
| BE3-eA3A + p53DD | GTG-GFP | BE3-eA3A | Lsg-GTG-GFP | p53DD |
| PE2 | GTG-GFP | pEF-PE2 | pegRNA-GTG-GFP-24 | |

| PE2 + p53DD | GTG-GFP | pEF-PE2 | pegRNA-GTG-GFP-24 | p53DD | |
|---|---|---|---|---|---|
| PE3 | GTG-GFP | pEF-PE2 | pegRNA-GTG-GFP-24 | Lsg-PE3-GTG-GFP | |
| PE3 + p53DD | GTG-GFP | pEF-PE2 | pegRNA-GTG-GFP-24 | Lsg-PE3-GTG-GFP | p53DD |

The GFP turn-on efficiency was evaluated at 24 h post-electroporation using flow cytometry. In order to get rid of the none-electroporated cells to minimize electroporation variation in different conditions, an addition of 1 µg of PUC57-puro was used in each electroporation reaction. Cells were selected with 0.5 µg/ml puromycin (InvivoGen # ant-pr-1) in Stemflex after 24 h post-electroporation for 1 day. The GFP turn-on efficiency was evaluated at 48 h post-electroporation using flow cytometry.

### A "H2B-tdTomato" endogenous reporter hESC line for evaluating CBE and PE efficiency

The sgRNA for CBE (BE3-FNLS were used here) was designed to convert C-to-T mutation and create a stop codon (TAG) on the H2B transgene. pegRNA for PE2 or PE3 was designed to directly insert a 3-bp "TGA" stop codon in the H2B-reading frame. To compare CBE and PE genome editing efficiency with or without p53DD in this endogenous reporter assay, seven different conditions were tested in the H1-H2B-tdTomato reporter line using a small electroporation reaction as described below. 1 µg of each plasmid was used.

Table of plasmids used to compare the PE editing efficiency using H2B-tdTomato reporter line.

| Condition | Plasmids combination | | | |
|---|---|---|---|---|
| Control | PUC57-puro | | | |
| BE3-FNLS | PUC57-puro | BE3-FNLS | Lsg-H2B-tdTomato | |
| BE3-FNLS + p53DD | PUC57-puro | BE3-FNLS | Lsg-H2B-tdTomato | p53DD |
| PE2 | PUC57-puro | pEF-PE2 | pegRNA-H2B-stop | |
| PE2 + p53DD | PUC57-puro | pEF-PE2 | pegRNA-H2B-stop | p53DD |
| PE3 | PUC57-puro | pEF-PE2 | pegRNA-H2B-stop | Lsg-H2B-nick |
| PE3 + p53DD | PUC57-puro | pEF-PE2 | pegRNA-H2B-stop | Lsg-H2B-nick | p53DD |

At 24 h post-electroporation, cells were selected with 0.5 µg/ml puromycin (InvivoGen #ant-pr-1) in Stemflex for 1 day to get rid of the none-electroporated cells. The tdTomato turn-off efficiency was evaluated at 48 h post-electroporation using flow cytometry. Cells were also collected for Miseq analysis.

### Apply PE to create PD or HGPS mutations

PegRNAs and nicking sgRNAs were designed using above described method for introducing GBA (c. 1226 A > G, p. N370S), LRRK2 (c. 6055 G > A, p. G2019S), and LMNA (c.1824 C > T; p.G608G) in a WT iPSC line (MSK-SRF001-iPSCs) respectively. PegRNAs and nicking sgRNAs were also designed to correct the mutations in a PD patient-iPSC line (756-iPSCs) carrying a heterozygous GBA (c. 1226 A > G, p. N370S) mutation, and in an HGPS patient-iPSC line (972-iPSCs) with a heterozygous LMNA (c.1824 C > T; p.G608G) mutation. To compare the PE editing efficiency with or without p53DD in WT and patients' iPSCs, the following conditions listed in the table below were tested:

Table of plasmids used to compare the PE editing efficiency for creating disease' isogenic lines.

| Parental line & purpose | Condition | Plasmids combination | | | |
|---|---|---|---|---|---|
| MSK-SRF001B-iPSCs-induce GBA (c. 1226 A > G) | PE2 | PUC57-puro | pEF-PE2 | pegRNA-GBA-10 | |
| | PE2 +p53DD | PUC57-puro | pEF-PE2 | pegRNA-GBA-10 | p53DD |
| | PE3 | PUC57-puro | pEF-PE2 | pegRNA-GBA-10 | Lsg-GBA-1 + 2 |
| | PE3 +p53DD | PUC57-puro | pEF-PE2 | pegRNA-GBA-10 | Lsg-GBA-1 + 2 | p53DD |
| MSK-SRF001B-iPSCs-induce LRRK2 (c. 6055 G > A) | PE2 | PUC57-puro | pEF-PE2 | pegRNA-LRRK2-3 | |
| | PE2+p53DD | PUC57-puro | pEF-PE2 | pegRNA-LRRK2-3 | p53DD |
| | PE3 | PUC57-puro | pEF-PE2 | pegRNA-LRRK2-3 | Lsg-LRRK2-1 |
| | PE3 +p53DD | PUC57-puro | pEF-PE2 | pegRNA-LRRK2-3 | Lsg-LRRK2-1 | p53DD |
| MSK-SRF001B-iPSCs-induce LMNA (c.1824C > T) | PE2 | PUC57-puro | pEF-PE2 | pegRNA-LMNA-1 | |
| | PE2 +p53DD | PUC57-puro | pEF-PE2 | pegRNA-LMNA-1 | p53DD |
| | PE3 | PUC57-puro | pEF-PE2 | pegRNA-LMNA-1 | Lsg-LMNA-1 + 2 |
| | PE3 +p53DD | PUC57-puro | pEF-PE2 | pegRNA-LMNA-1 | Lsg-LMNA-1 + 2 | p53DD |
| 756-iPSCs-correct GBA (c. 1226 A > G) | PE2 | PUC57-puro | pEF-PE2 | pegRNA-GBA-4 | |
| | PE2 +p53DD | PUC57-puro | pEF-PE2 | pegRNA-GBA-4 | p53DD |
| | PE3 | PUC57-puro | pEF-PE2 | pegRNA-GBA-4 | Lsg-GBA-1 + 2 |
| | PE3 +p53DD | PUC57-puro | pEF-PE2 | pegRNA-GBA-4 | Lsg-GBA-1 + 2 | p53DD |
| 972-iPSCs-correct LMNA (c.1824C > T) | PE2 | PUC57-puro | pEF-PE2 | pegRNA-LMNA-CR-2-5 | |
| | PE2 +p53DD | PUC57-puro | pEF-PE2 | pegRNA-LMNA-CR-2-5 | p53DD |
| | PE3 | PUC57-puro | pEF-PE2 | pegRNA-LMNA-CR-2-5 | Lsg-LMNA-1 + 2 |
| | PE3 +p53DD | PUC57-puro | pEF-PE2 | pegRNA-LMNA-CR-2-5 | Lsg-LMNA-1 + 2 | p53DD |

At 24 h post-electroporation, cells were selected with 0.5 µg/ml puromycin (InvivoGen #ant-pr-1) in Stemflex 24 h post-electroporation for 1 day to get rid of the none-electroporated cells. At 48 h post-electroporation, cells were collected for Miseq analysis. Single-cell clones were also isolated and expanded from PE3 + p53DD condition for mutation screening with Sanger sequencing, and further characterizations.

### Miseq analysis

Cells at 48 h post-electroporation at each condition were lysed using Lysis Solution for Blood (Millipore Sigma L3289), followed by

neutralization with Neutralization Solution for Blood (Millipore Sigma SRE0087). A pair of primers were designed to amplify the editing area for PCR (amplicon size was ~300-450 bp), using KOD Xtreme Hot Start DNA Polymerase ultrahigh fidelity (EMD Millipore) from cell lysate. PCR products were purified by gel extraction, and then submitted for Amplicon sequencing (Illumina MiSeq system, Amplicon EZ service from Genewiz). ~50,000–100,000 total reads per sample were collected. Alignment of amplicon sequences to a reference sequence was performed using CRISPResso2[45]: (https://crispresso.pinellolab.partners.org/submission). For all editing quantifications, we set these parameters: Minimum average read quality (phred33 scale) was >30; Exclude 15 bp from the left side and 15 bp from the right side of the amplicon sequence for the quantification of the mutations. The Minimum homology for alignment to an amplicon was set >60%. sgRNA and pegRNA targets were provided and the quantification window was set as −3. To quantify insertion or deletion edits, CRISPResso2 was run in HDR mode using the sequence with desired insertion or deletion editing as the reference sequence. Editing yield was calculated as a percentage of perfect HDR-aligned reads/total aligned reads. For quantification of point mutation editing, CRISPResso2 was run in standard Cas9 mode. Editing yield was calculated as the percentage of reads containing the edit divided by total reads. For quantification of mutation correction in patients' iPSCs that originally carried a heterozygous point mutation, the correction efficiency was calculated as: (percentage of a number of reads with the corrected nucleotide - percentage of the number of reads with the mutant nucleotide)/2. For quantification of potential indels at the sgRNA nicking site, the indel efficiency was calculated as: a percentage of (number of indicated indel-containing reads) / (total reads). PCR primers used in Miseq are listed in Supplementary table 4.

### Single-cell clone screening by sanger sequencing

Single-cell clone generation, PCR, and Sanger sequencing were performed following our previously reported protocol[38]. Briefly, 3–4 days post-electroporation, PE3 + p53DD edited WT iPSCs or patients' iPSCs were dissociated into single cells by Accutase (Innovative Cell Tech. AT104), and single-cell clones were allowed to form in 96-well plates. 10-12 days later, individual colonies were picked, and mechanically disaggregated into two copies. One copy was lysed for PCR and Sanger sequencing, while the other copy was maintained in cell culture. Specifically, KOD Xtreme Hot Start DNA Polymerase ultrahigh fidelity (EMD Millipore) or Q5 Hot Start High-Fidelity 2x Master Mix (New England BioLabs) was used for PCR from the single-cell clone lysis using the same PCR primer sets as used for Miseq, followed by treatment with ExoSAP-IT PCR Product Cleanup Reagent (Thermo Fisher) for Sanger sequencing at Eton Biosciences. PCR primers used in Sanger sequencing are listed in Supplementary table 4.

### p53DD integration detection

Single-cell clones were passaged 2–4 times in 6-well plates to achieve enough cell numbers for stocking and genomic DNA extraction. Genomic DNA was extracted using DNeasy Blood & Tissue Kits (Qiagen 69504). A pair of specific primers were designed (PCR-forward STCLM0382F: 5′ GGGCGTAAACGCTTCGAGAT 3′, and PCR reverser STCLM0383R: CAAGGTCACCAGACAGAGATGCT) to amplify a part of the p53DD construct (PCR size: 1027 bp). A pair of LMNA gene primers were used as the PCR reaction control for evaluating the quality of the genomic DNA and the PCR reactions (PCR-forward STCLM0253F: 5′ TGGGCACAGAACCACACCTTC 3′, and PCR reverser STCLM0254R: 5′ AGACAAAGCAGAGACAACTCACCT 3′) (PCR size: 414 bp). A positive control (p53DD plasmid), and negative control (cells without p53DD electroporation) were performed at the same time for each experiment.

### Flow cytometry analysis

Prior to FACS analysis, cells were detached using Accutase (Innovative Cell Tech. AT104) at 37 °C for 10 min, followed by the addition of Stemflex Medium and centrifugation at 120 g for 3 min. Cell pellets were resuspended in 300 μl Stemflex, filtered through a Falcon 5 mL Round Bottom Polystyrene Test Tube with Cell Strainer Snap Cap (Fisher Scientific 352235), and kept on ice. Cells were then analyzed for either FITC-A (350 V), DsRed (350 V), and/or mCherry (400 V) signals using a BD FACSAria III (BD Bioscience). Data analysis was done using BD FACSDiva Software version 8.0. (BD Bioscience). The gating strategy is exemplified in Supplementary Fig. 18.

### qPCR to detect CDKN1A expression

seven different vector combinations including GTG-GFP vector alone, GTG-GFP vector with CBE tools, GTG-GFP vector with CBE tools and p53DD, GTG-GFP vector with PE2 tools, GTG-GFP vector with PE2 tools and p53DD, GTG-GFP vector with PE3 tools, GTG-GFP vector with PE3 tools and p53DD were electroporated to H1 hESCs. ~18 h post-electroporation, cells were collected for RNA extraction using the RNeasy Mini Kit (Qiagen, 74104), and cDNA was synthesized with the SuperScript VILO Master Mix (Invitrogen, 11755050). qPCR primers of CDKN1A (Forward: 5′-GGAGACTCTCAGGGTCGAAAAC-3′, and Reverse: 5′-TTCCTGTGGGCGGATTAGG-3′), and GAPDH control (Forward: 5′-GGCTGAGAACGGGAAGCTT-3′, and Reverse: 5′-AGGGATCTCGCTCCTGGAA-3′) were used, and qPCR was performed with QuantStudio 5 (Applied Biosystems).

### RNA-sequencing and variant analysis

To assess transcriptome-wide off-target induced by CBE, H1-SOX2-H2B-tdTomato cells were transfected with CBE with or without P53DD to introduce a stop codon in the H2B reading frame. Total RNA was extracted from the sorted tdTomato negative cells after 48 hours of nucleofection by using RNeasy Mini Kit (Qiagen). RNA integrity and quantity were then determined by Alilent TapeStatin 4200 (Agilent Technologies) and Qubit 2.0 instrument respectively. RNA library was prepared using NEBNext Ultra RNA Library Prep Kit (New England Biolabs) for Illumina following the manufacturer's instructions and underwent 2 × 150-bp sequenced on an Illumina HiSeq System by GENEWIZ.

RNA-seq data alignment analysis was performed by MSK Bioinformatic Core. Fastq files were mapped to the targeted genome using the STAR aligner[46] version 2.5.0a (Linux_x86_64). Two-pass alignment was used in which the reads are mapped twice[47]. PICARD tools were used for post-processing of the output SAM files to add read groups and convert SAM files to a compressed BAM format. RNA base-editing variants were called using standard GATK. To quantify C-to-U changes, variants were further filtered by comparison with an unedited control sample as follows: (1) The control sample has to be either a "C" or a "G". (2) The base frequency has to be >99%; i.e., of all the reads mapping to the site, only 1% can be an "error". (3) The mutation has to be C > T or G > A. (4) The control sample has to have a depth >3.

### WGS and data analysis

Genomic DNA from hPSCs was isolated using DNeasy Blood & Tissue Kit (QIAGEN) according to the manufacturer's instructions. DNA libraries were prepared with standard Illumina protocols and were sequenced with Illumina Hiseq 4000 platform using 2 × 150 bp paired-end configuration by GENEWIZ. Base calls and quality scores were stored in.bcl files which were then converted into fastq files by using HiSeq Analysis Software (HAS) v2.2 suite. De-multiplexing was performed according to barcodes for the samples. For data analysis, sequence reads were mapped against the human reference genome (NCBI GRCh38) using Issac Aligner and the identified duplicate reads were removed from downstream analysis. The Issac Variant Caller was applied to detect SNVs and small indels up to 50 bp in the samples compared to the reference genome sequence[48].

For both CBE and PE edited samples, eight additional filters were performed to obtain high-confidence variant calls: 1. IndelConflict to remove locus which in region with conflicting indel calls; 2. SiteConflict to exclude sites with an overlapping indel call; 3. LowGOX to filter out locus with GQX (genotyping quality score) less than 30 or not present; 4. HighDPFRatio to remove a fraction of base calls at a site greater than 0.4; 5. HighSNVSB to filter out SNVs with strand bias value exceeds 10; 6, 7. HighDepth and LowDepth to remove locus with a depth >3× the mean chromosome depth and locus depth below 3; 8. PloidyConflict to filter out genotype call from a variant caller not consistent with chromosome ploidy. For the sorted tdTomato negative cells with CBE editing, variants that were also present in the parental H1-SOX2-H2B-tdTomato cells were filtered out to retain only de novo variants generated by the editing tool.

### Reporting summary
Further information on research design is available in the Nature Research Reporting Summary linked to this article.

### Data availability
Human reference genome (GRCh38) was downloaded from GenBank. The Amplicon sequencing and WGS data generated in this study have been deposited in the NCBI BioProject ID: PRJNA812517. All data for the graphs presented in this study are provided in the Source Data File.

### Code availability
The Benchling CRISPR Design website was used for designing sgRNA or pegRNA spacer and is available at (https://www.benchling.com/#). CRISPResso2 website was used for the analysis of Amplicon sequencing (Miseq) to determine the PE and CBE editing efficiency and is available at (https://crispresso.pinellolab.partners.org/submission).

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

## Acknowledgements

This work was supported by NIH/NCI Cancer Center Support Grant P30 CA008748 from Memorial Sloan Kettering. The work was also supported by a core facility grant of the Starr Foundation through the Tri-Institutional Stem Cell Initiative, and by the Contract C029153 from the New York State's stem cell funding agency (NYSTEM), and by R35GM145260 to X.Z. We thank the MSK Bioinformatic Core for transcriptome-wide SNV analysis from the RNA-seq data and WGS data.

## Author contributions

T.Z., L.S., M.L., Y.W. designed the project. M.L. A.Z., Y.W., M.S., M.B., T.Z. performed the experiments. M.L. A.Z., Y.W., T.Z. analyzed data. T.Z., L.S., M.L., A.Z., Y.W., X.Z. wrote the manuscript.

## Competing interests

L.S. is a scientific co-founder and consultant of BlueRock Therapeutics. There are no competing interests for any of the other authors.

## Additional information

**Correspondence and requests** for materials should be addressed to Lorenz Studer or Ting Zhou.

