## [Peer Review File · Nature Communications]

Transient inhibition of P53 enhances prime editing and cytosine base editing efficiencies in human pluripotent stem cellsREVIEWER COMMENTS

Reviewer #1 (Remarks to the Author):

In the manuscript entitled "Transient inhibition of P53 enhances prime editing and base editing efficiencies in human pluripotent stem cells, Li et al. describe a very interesting finding: expression of p53 dominant negative protein from an EBV-based episomal vector increases efficiency of prime editing and base editing.

The work is of practical utility, and also potentially quite important for understanding the mechanism of DNA repair and its interaction with cell cycle progression. However, in its present form, the manuscript is too dependent on a single transfection method (electroporation) and a single vector construct that co-expresses a multifunctional viral protein, EBNA-1.

It remains possible that electroporation damages DNA and prevents cell cycle progression in a p53 dependent manner. Failure to enter DNA replication may then affect the rate of formation of alleles with the targeted edit. Also, it is not clear which described effects are due to p53DD and which due to EBNA-1, and/or their combination.

Regardless of the mechanism, the finding is of practical importance. However, for publication in a Nature family journal, the authors should at least isolate the effect to a specific input variable. That is, does the finding depend on electroporation? and can expression of EBNA-1 alone affect editing efficacy, or would inhibition of p53 by some other mechanism also improve base editing and prime editing.

Other points:

- The efficiency of PE2 is not always improved by the co-electroporation. Could this be caused by different levels of expression of the target genes? Polymerases that encounter nicks or DNA damaged loci may trigger strong responses.

- the authors need to more clearly explain their episomal expression system to the general audience.

- the authors should discuss the effect of electroporation on the cells. Does this alone induce p21?

- Does inhibition of the cell cycle by CDK4-6 inhibitors also decrease PE and BE activity?

Reviewer #2 (Remarks to the Author):

In the manuscript “Transient inhibition of p53 enhances prime editing and base editing efficiencies in human pluripotent stem cells”, Li et al. convincingly demonstrated that transient expression of a p53 carboxy-terminal dominant-negative fragment (p53DD) can increase prime editing and cytosine base editing efficiencies by ~100-200% in human iPS cells. The backdrop of this study is that Cas9-mediated double-strand DNA breaks (DSBs) are known to induce p53-dependent cell cycle arrest and cell death; hence, p53 inhibition has been shown to improve CRISPR-Cas9 genome editing efficiencies in untransformed cells (Ref #8 and #9 in the manuscript). In the current report, the authors show that P53DD increased both the intended edit and the on-target indel frequencies with a base editor. P53DD also increased the intended edit frequencies with the prime editor. However, the on-target indel frequencies with the PE method were mostly very low and they appeared unaffected by p53DD. In sum, these findings are significant and may change future practices for using these genome editing tools in human iPS cells and beyond. However, the study still has some notable shortcomings as outlined below.

1. It is unclear why p53 inhibition could increase PE and BE edit frequencies. A better mechanistic understanding should be provided. Based on the results shown in SI Fig 9, Li et al. suggested that the p53-p21 axis might be activated in CBE and PE3 edited cells. Is this due to the single-strand DNA nick or is it driven by Cas9 binding to DNA (Ref #17 in the manuscript)? The authors may use a base editor fused to dCas9 to see if it can induce p21 transcription and if P53DD can increase its base editing efficiencies.
2. Since p53 inhibition via plasmid may result in unwanted DNA integration as shown in this study, why not use p53 siRNA or chemical inhibition of p53 activities?
3. The authors have not investigated the role of p53 inhibition on potential off-target effects of these gene editing tools. For example, CBE has been shown to cause deamination of cytosines in DNA and RNA independent of sequence-specified CBE binding (Zuo, et al. Science 2019; Kim, et al. Nature Biotechnology 2017; Grunewald, et al. Nature 2019). How does p53 inhibition impact these off target activities?

Other comments:

1. For Miseq data analysis, please specify which mode of CRISPResso2 was used and the specific parameters used.
2. In the sentence from Line 158 to Line 162, the authors concluded that the results suggest low “off-target” potential of PE. Indels at pegRNA nicking sites are not considered as “off-target” activities. I would suggest change “off-target” to “byproduct”.

Reviewer #3 (Remarks to the Author):

Comments/Suggestions

1. BE can mediate direct conversion of C-to-T or A-to-G and PE can introduce all types of SNP changes, precise deletion or insertions at the specific site of a genome. In this work, the author described "Transient inhibition of P53 enhances prime editing and base editing efficiencies in human pluripotent stem cells". But the author didn't verify the effect of P53DD on PE-mediated mutation/deletion and ABE system. These data should be appended at least in endogenous genome sites, otherwise it was incomplete.

2. For Supplementary Fig. 5b, c, the author described "the results suggest P53DD increases PE editing efficiencies for "CTT" or "Loxp" insertion at HEK3 locus, without notably increasing indels frequencies". But there was obvious increase of indels in "CTT" insertion site of PE3. Also, two endogenous sites were too few to conclude the difference of indels accurately. To clearly explain the effect of indel by P53DD, more endogenous genome sites should be tested.

3. In this work, the author isolated and expanded many single-cell clones to detect indels and other bystander products from the PE3+P53DD editing samples. In fact, it may be of no significance. To comprehensively assess the effect of PE3 with or without P53DD, besides Sanger sequencing and Miseq on target sites, genome-wide sequencing analysis of single-cell clone was essential.

Reviewers' comments:

Reviewer #1. In the manuscript entitled "Transient inhibition of p53 enhances prime editing and base editing efficiencies in human pluripotent stem cells, Li et al. describe a very interesting finding: expression of p53 dominant negative protein from an EBV-based episomal vector increases efficiency of prime editing and base editing.

The work is of practical utility, and also potentially quite important for understanding the mechanism of DNA repair and its interaction with cell cycle progression. However, in its present form, the manuscript is too dependent on a single transfection method (electroporation) and a single vector construct that co-expresses a multifunctional viral protein, EBNA-1. It remains possible that electroporation damages DNA and prevents cell cycle progression in a p53 dependent manner. Failure to enter DNA replication may then affect the rate of formation of alleles with the targeted edit. Also, it is not clear which described effects are due to p53DD and which due to EBNA-1, and/or their combination.

We appreciate the reviewer's very positive comments on our work, its practical importance, and its potential use for understanding DNA repair in relationship with the cell cycle regulation.

The reviewer raised two important questions here. First, the review wondered whether the effect of p53DD is dependent on electroporation. As hPSCs is hard to transfect chemically, electroporation is the most widely adopted delivery method for hPSCs^{1,2}. We have already shown that the electroporation alone did not increase the p21 gene expression (**Supplementary Fig. 13**), thus unlikely causing a p53-mediated cell cycle arrest. To test additional delivery methods and evaluate their effects with p53DD, we used Lipofectamine Stem Transfection Reagent (LipoStem)³, to co-transfect PE2, PE3, or CBE with or without p53DD into H1-H2B-tdTomato endogenous reporter cells. Despite the lower editing efficiency with LipoStem compared to electroporation, p53DD still increased ~2x of PE or CBE editing efficiency, which is consistent with the results we obtained with electroporation (**new Supplementary fig. 6**, attached below). These results suggest that p53DD can enhance PE and CBE editing efficiency in a delivery method-independent manner. We have added these new results in the manuscript on **Page 8 from Line 15 to 22**.

The second question relates to the p53DD episomal vector (pCE-p53DD) that we used in the study. The reviewer is correct that the construct carries two functional components: a p53DD (p53 dominant-negative) fragment and an EBNA-1 (which is a multifunctional viral protein) fragment. To address the question whether the effects of the p53DD construct is derived from the p53DD fragment or the EBNA-1 fragment, we modified the original pCE-p53DD construct to generate a new EBNA-1 expression construct (EBNA-1), and a new p53DD-only expression construct (p53DD-only) (**new Supplementary Fig. 7a**, attached below). Importantly, both new constructs and the original pCE-p53DD are all under the control of an identical CAG-promoter (**new Supplementary Fig. 7a**, attached below). We then tested the EBNA-1 and p53DD-only constructs in the PE and CBE co-electroporation experiments, using the original pCE-p53DD as control. We found co-electroporation of p53DD-only, but not the EBNA-1 expression construct increased the PE and CBE editing efficiency, with highly comparable results to the pCE-p53DD control (**new Supplementary Fig. 7b-e**, attached below). These data demonstrate that EBNA-1 alone cannot increase the PE or CBE editing efficiency. Either p53DD episomal vector or a simple "p53DD-only" expression vector can enhance the PE or CBE editing efficiency to the similar level. We have added these new results in the manuscript on **Page 9 from Line 1 to 14**.

Regardless of the mechanism, the finding is of practical importance. However, for publication in a Nature family journal, the authors should at least isolate the effect to a specific input variable. That is, does the

finding depend on electroporation? and can expression of EBNA-1 alone affect editing efficacy, or would inhibition of p53 by some other mechanism also improve base editing and prime editing.

We appreciate the reviewer's suggestion to test inhibition of p53 by some alternative mechanisms.

First, we tested p53 siRNA, and found co-electroporation of p53 siRNA did not increase PE or CBE editing efficiency (**new Supplementary Fig. 8a-d**, attached below), which is likely due to the different kinetics between genome editing and RNA silencing⁴, and that p53 is mainly regulated at the protein level^{5,6}.

Second, we tested p-nitro-Pifithrin- α (a cell-permeable PFT- α), a widely used small-molecule p53 inhibitor⁷, and we found that only high dose of p-nitro PFT- α (10 μ M) can increase the CBE, but not PE2 or PE3 editing efficiency (**new Supplementary Fig. 8a, e-g**, attached below). A recent study discovered that PFT- α displays other p53-independent activity in cells and fails to prevent the p53 effects on cell cycle and apoptosis in some cases. The report further suggests caution of using PFT- α to study p53-dependent processes⁸. Our data shows that p53 siRNA or PFT- α cannot replace p53DD to enhance the PE or CBE editing efficiency in hPSCs.

Lastly, we tested co-electroporation of a MDM2 expressing vector⁹ with PE or CBE editing tools. MDM2 is the main E3 ligase for p53 ubiquitination in cells, which mediates poly-ubiquitination of p53 and directs it for proteasomal degradation^{6,10,11}. We found co-electroporation of MDM2 can mimic the effect of p53DD in hPSCs by significantly improving the PE or CBE editing efficiency (**new Supplementary Fig. 8h-i**, attached below).

We have added these new results in the manuscript on **Page 9 Line 16-25**, and **Page 10 Line 1-7**.

Supplementary Fig. 6

New Supplementary Figure 6. p53DD increased the PE and CBE editing efficiencies via LipoStem delivery method.

a, Schematics of evaluating the effects of adding the p53DD plasmid into the PE and CBE editing tool kits respectively via LipoStem co-transfection, in “H1-H2B-tdTomato” endogenous reporter cells. The % of tdTomato negative cells was evaluated at 48h post-transfection as the read out. LipoStem transfection reagent were purchased from Thermo Fisher Scientific (STEM00008).

b, c, Flow cytometry analysis (**b**), and the quantification of tdTomato negative population % at 48 h post-electroporation with CBE tools (sgRNA and BE3-FNLS), with or without p53DD. Unedited H1-H2B-tdTomato cell line was used as a control (**c**). $n = 3$ independent LipoStem transfection reactions for each condition. Values presented as mean \pm S.D. p values were calculated by ordinary one-way ANOVA were * $p < 0.05$; *** $p < 0.0001$.

d-f, Flow cytometry analysis (**d**), and quantification of the tdTomato negative population % at 48 h post-electroporation with PE2 or PE3 tools, with or without p53DD. Unedited H1-H2B-tdTomato cell line was used as a control (**e, f**). $n = 3$ independent LipoStem transfection reactions for each condition. Values presented as mean \pm S.D. p values were calculated by ordinary one-way ANOVA were **** $p < 0.0001$.

New Supplementary Figure 7. p53DD-only fragment, but not the EBNA1 fragment, increased the PE and CBE editing efficiencies.

a, A scheme showing the re-construction of the original pCE-p53DD episomal vector. To generate a vector without p53DD but only carrying the EBNA-1 expression fragment, EcoRI was used to digest out the p53DD fragment of pCE-p53DD vector, following by the vector self-ligation to create the new EBNA-1 expression vector (EBNA-1). To generate a vector that carries only p53DD expression fragment, NheI and AvrII was used to digest out the EBNA-1 fragment of pCE-p53DD vector, following by the vector self-ligation to create the new p53DD-only expression vector (p53DD-only).

b, c, Flow cytometry analysis (**b**), and the quantification of tdTomato negative population % at 48 h post-electroporation with CBE tools (sgRNA and BE3-FNLS), CBE tools with EBNA-1 or p53DD-only or pCE-p53DD (**c**). Unedited H1-SOX2-tdTomato reporter cells was used as gating control, CBE only was used as the editing control, CBE with original pCE-p53DD was used as positive control. $n = 3$ independent electroporation reactions for each condition. Values presented as mean \pm S.D. p values were calculated by ordinary one-way ANOVA were **** $p < 0.0001$. ns. indicates non-significant difference.

d, e, Flow cytometry analysis (**d**), and the quantification of tdTomato negative population % at 48 h post-electroporation with PE2 or PE3 tools, PE2 or PE3 tools with EBNA-1 or p53DD-only or pCE-p53DD (**e**). Unedited H1-H2B-tdTomato cell line was

used as gating control, PE2 or PE3 only were used as the editing control, PE2 or PE3 with original pCE-p53DD was used as positive control. $n=3$ independent electroporation reactions for each condition. Values are presented as mean \pm S.D. p values were calculated by ordinary one-way ANOVA. **** $p < 0.0001$. ns. indicates non-significant difference.

Supplementary Fig.8

New Supplementary Figure 8 (a-g). p53 siRNA or p53 inhibitor p-nitro PFT-α cannot replace p53DD to enhance the PE or CBE editing efficiency in hPSCs.

a. Schematics of evaluating the effects of co-electroporation of control or P53 siRNA, or adding p53 chemical inhibitor p-nitro-Pifithrin-α into the PE and CBE editing tool sets respectively, based on “H2B-tdTomato” endogenous reporter assay. tdTomato turn-off rate at 48 h post-electroporation was as the read out. Control siRNA (CST, 6568) or p53 siRNA (CST, 6562) were purchased from Cell Signaling Technology. p-nitro-Pifithrin-α were purchased from Cayman Chemicals (16209).

b-d. The quantification of tdTomato negative population edited in the conditions of CBE (BE3-FNLS) (**b**), PE2 (**c**) or PE3 (**d**) tools, with control siRNA or p53 siRNA, in flow cytometry analysis. 100nM control siRNA or 100nM p53 siRNA was used in each electroporation reaction. $n=3$ independent experiments for each condition. Values presented as mean \pm S.D. ns. indicates non-significant difference.

e-g. The quantification of tdTomato negative population edited in the conditions of CBE (BE3-FNLS) (**e**), PE2 (**f**) or PE3 (**g**) tools, with indicated doses of p-nitro-Pifithrin-α, in flow cytometry analysis. Upon electroporation, the cells were split to 96-well plates in the medium with of 3 μM or 10 μM of p-nitro-Pifithrin-α, or 1000x DMSO. $n=3$ wells total for CBE, $n=9$ wells total for PE2, $n=12$ wells total for PE3, from 3 independent experiments for each condition. Values presented as mean \pm S.D. p values were calculated by ordinary one-way ANOVA were * $p < 0.05$. ns. indicates non-significant difference.

New Supplementary Figure 8 (h-l). Co-electroporation of MDM2 expression vector significantly increases the PE and CBE editing efficiencies.

h. Schematics of evaluating the effects of co-electroporation of MDM2 expression vector (pcDNA3 MDM2 WT (Plasmid #16233), with the PE or CBE editing tools based on “H2B-tdTomato” endogenous reporter assay. tdTomato turn-off rate at 48 h post-electroporation was as the read out.

i-l, Flow cytometry analysis (i), and quantification of the tdTomato negative population % at 48 h post-electroporation in the editing conditions of CBE(BE3-FNLS) (j), or PE2 (k), or PE3 (l) tools, with or without MDM2. n= 4 independent experiments for CBE or PE3 editing conditions. n= 3 independent experiments for PE2 editing conditions. Values presented as mean ± S.D. p values calculated by ordinary one-way ANOVA were **p < 0.01, ***p < 0.001, and ****p < 0.0001.

Other points:

-The efficiency of PE2 is not always improved by the co-electroporation. Could this be caused by different levels of expression of the target genes? Polymerases that encounter nicks or DNA damaged loci may trigger strong responses.

Our data showed that p53DD improved PE2 editing efficiency in both highly expressed (SOX2-tdTomato) and silenced (HEK3) loci in hPSCs, thus the RNA polymerase status alone is unlikely to determine editing efficiency. While the reason for a lack of effect of p53DD on PE2 editing at some disease-relevant genes tested here is not fully clear, we suspect that chromatin environment at these loci may disfavor editing. We now added this point on **Page 12, Line 14-16** as an idea to be examined in future studies.

- the authors need to more clearly explain their episomal expression system to the general audience.

As proposed by the reviewer, we now have added the description of the episomal expression system in the manuscript to explain the key features in simple terms on **Page 14 Line 15-17**, as follows:

“The episomal expression system contains OriP/EBNA1 components of Epstein-Barr virus, which enables relatively high and long-term gene expression in the host cells, while still allowing for gradual loss of vector from proliferating cells¹².”

- the authors should discuss the effect of electroporation on the cells. Does this alone induce p21?

We showed that electroporation alone does not induce *CDKN1A* (encoding p21) (original SI Fig 9, the **new Supplementary Fig. 13**).

- Does inhibition of the cell cycle by CDK4-6 inhibitors also decrease PE and BE activity?

We performed new PE and CBE editing experiments with a CDK4/6 inhibitor (palbociclib)¹³ or a DNA synthesis inhibitor (thymidine) to arrest cell cycle at the G1/S phase transition¹⁴. Flow cytometry analysis showed that both inhibitors decreased PE2 or PE3 editing efficiencies in a dose-dependent manner (**new Supplementary Fig. 14 a-c**, attached below). They also decreased CBE editing efficiency at their effective doses (**new Supplementary Fig. 14 d, e**, attached below). Survival of hPSCs was not changed under the dosage used. These results suggest that interference of cell cycle progression by these inhibitors actually decreases the PE or CBE editing activity.

We have included these new results and incorporated in the “Discussion” section **on Page 15 Line 20-24**.

New Supplementary Figure 14. CDK4-6 inhibitor (Palbociclib), or a DNA synthesis inhibitor (Thymidine) treatment decreased PE and CBE editing efficiencies in hPSCs.

a-e, Flow cytometry analysis (**a**), and the quantification of tdTomato negative population % at 48 h post-electroporation with PE2 (**a,b**) or PE3 (**a,c**) or CBE (BE3-FNLS) (**d,e**) editing tools, with or without indicated doses of Palbociclib and Thymidine. Palbociclib were purchased from Sigma (T1895). Thymidine were purchased from Sigma (PZ0383). Upon electroporation, the cells were split to 96-well plates in the medium with of 5 μM or 10 μM or 20 μM of Palbociclib, 400 μM or 800 μM of Thymidine, or 1000x DMSO. For PE2 and PE3 editing experiments, $n=3$ wells total for 5 μM Palbociclib or 20 μM Palbociclib treatment; $n=6$ wells total for all other condition treatment, from 3 independent experiments. For CBE editing experiments, $n=3$ wells total for each condition from 3 independent experiments. Unedited H1-SOX2-tdTomato reporter cells was used as gating control, PE2 or PE3 or CBE with DMSO condition was used as the editing control. Values presented as mean \pm S.D. p values calculated by ordinary one-way ANOVA were * $p < 0.05$, ** $p < 0.01$, *** $p < 0.001$, and **** $p < 0.0001$. ns. non-significant difference.

Reviewer #2. In the manuscript “Transient inhibition of p53 enhances prime editing and base editing efficiencies in human pluripotent stem cells”, Li et al. convincingly demonstrated that transient expression of a p53 carboxy-terminal dominant-negative fragment (p53DD) can increase prime editing and cytosine base editing efficiencies by ~100-200% in human iPS cells. The backdrop of this study is that Cas9-mediated double-strand DNA breaks (DSBs) are known to induce p53-dependent cell cycle arrest and cell death; hence, p53 inhibition has been shown to improve CRISPR-Cas9 genome editing efficiencies in untransformed cells (Ref #8 and #9 in the manuscript). In the current report, the authors show that p53DD increased both the intended edit and the on-target indel frequencies with a base editor. p53DD also increased the intended edit frequencies with the prime editor. However, the on-target indel frequencies with the PE method were mostly very low and they appeared unaffected by p53DD. In sum, these findings are significant and may change future practices for using these genome editing tools in human iPS cells and beyond. However, the study still has some notable shortcomings as outlined below.

We thank the reviewer for the very positive comments that our data are convincing and significant and that this work may change the future practice of using PE and CBE tools in human iPSC cells.

1. It is unclear why p53 inhibition could increase PE and BE edit frequencies. A better mechanistic understanding should be provided. Based on the results shown in SI Fig 9, Li et al. suggested that the p53-p21 axis might be activated in CBE and PE3 edited cells. Is this due to the ssDNA nick or is it driven by Cas9 binding to DNA (Ref #17 in the manuscript)? The authors may use a base editor fused to dCas9 to see if it can induce p21 transcription and if p53DD can increase its base editing efficiencies.

Our data showing increased levels of p21 expression in CBE and PE3 edited cells is compatible with activation of the p53-p21 axis in these cells (original SI Fig 9, **new Supplementary Fig. 13**). This result suggests a potential model wherein transient inhibition of p53-mediated cell cycle arrest and/or apoptosis can improve PE and CBE editing efficiencies in hPSCs. Further supporting this theory, we now provide new data to show that arresting cells at the G1/S transition by a CDK4/6 inhibitor (palbociclib)¹³ or a DNA synthesis inhibitor (thymidine)¹⁴ has the opposite effects as p53DD (**new Supplementary Fig. 14**, attached above). We have included these new results and incorporated in the “Discussion” section **on Page 15 Line 20-24**.

We also performed the suggested experiment to test a dCas9-based base editor. We re-constructed the CBE vector (eA3A-BE3)¹⁵ by fusing the eA3A domain and dCas9 to generate the dCas9-CBE (eA3A-BE2) editing tool (**new Supplementary Fig.15a**, attached below). We found that electroporation of eA3A-BE2 editing tools did not induce p21 transcription and that p53DD did not increase eA3A-BE2 editing efficiency (**Supplementary Fig. 15b-d**; attached below). These results indicated that DNA nicking, but not DNA binding per se, by Cas9 causes the induction of p53-p21 axis. It is worthwhile to note that given that BE2 editing efficiency is generally low in hPSCs, further testing using other editing system may help to better understand the mechanism by which p53 inhibition can affect gene editing efficiency in hPSCs.

In summary, the self-consistent data described above support the action model that co-electroporation of p53DD removes the road block of cell cycle inhibition induced by PE or CBE editing tools, and allows gene editing to proceed more efficiently in hPSCs (**new Supplementary Fig.16**, attached below). We have included new results, the proposed mechanistic model and the discussion on **Page 16 Line 3-14**.

New Supplementary Figure 15. p53DD did not increase the editing efficiency of dCas9-CBE (eA3A-BE2) in hPSCs.

a, A scheme showing the re-constitution of the CBE vector (eA3A-BE3) by replacing the Cas9n to dCas9 to create dCas9-CBE (eA3A-BE2) editing tool.

b, qPCR quantification of CDKN1A (p21 gene) expression in GFP expression vector electroporated cells or eA3A-BE2 edited cells. RNA sample were collected at 48 h post-electroporation. The relative expression value of each condition was normalized to non-electroporated H1-H2B-tdTomato reporter hESCs. $n = 3$ independent electroporation reactions for each editing condition. Values presented as mean \pm S.D. p values were calculated by one-way ANOVA test, and indicate non-significant difference.

c, d, Flow cytometry analysis (**c**), and the quantification of tdTomato negative population % at 48 h post-electroporation with eA3A-BE2 editing tool, with or without p53DD (**d**). $n = 3$ independent electroporation reactions for each condition. Values presented as mean \pm S.D. p values were calculated by ordinary one-way ANOVA, and indicate non-significant difference.

New Supplementary Figure 16. Proposed Mechanism of p53DD to enhance PE and CBE editing efficiency.

In response to stresses, p53 functions as a tetramer to regulate downstream gene transcription¹⁶. The p53DD acts as the dominant negative format of WT p53, which carries the tetramer domain and C-terminal domain, but lacks of the DNA-binding domain¹⁷. p53DD protein is capable of forming mixed tetramers with WT p53 proteins, and prevent the binding of the p53 tetramer to DNA^{18,19} and consequently gene transcription, e.g. p21 gene expression. The consequent release of potential cell cycle arrest can explain how p53DD improves PE and CBE editing efficiency in hPSCs.

2. Since p53 inhibition via plasmid may result in unwanted DNA integration as shown in this study, why not use p53 siRNA or chemical inhibition of p53 activities?

This is a good point, and a similar point was raised by Reviewer #1. As detailed in our response to Reviewer #1, we tested three additional strategies for p53 inhibition.

First, we tested p53 siRNA, and found co-electroporation of p53 siRNA did not increase PE or CBE editing efficiency (**new Supplementary Fig. 8a-d**), which is likely due to the different kinetics between genome editing and RNA silencing⁴, and that p53 is mainly regulated at the protein level^{5,6}.

Second, we tested p-nitro-Pifithrin- α (a cell-permeable PFT- α), a widely used small-molecule p53 inhibitor⁷, and we found that only high dose of p-nitro PFT- α (10 μ M) can increase the CBE, but not PE2 or PE3 editing efficiency (**new Supplementary Fig. 8a, e-g**). A recent study discovered that PFT- α displays other p53-independent activity in cells and fails to prevent the p53 effects on cell cycle and apoptosis in some cases. The report further suggests caution of using PFT- α to study p53-dependent processes⁸. Our data shows that p53 siRNA or PFT- α cannot replace p53DD to enhance the PE or CBE editing efficiency in hPSCs.

Lastly, we tested co-electroporation of a MDM2 expressing vector⁹ with PE or CBE editing tools. MDM2 is the main E3 ligase for p53 ubiquitination in cells, which mediates poly-ubiquitination of p53 and directs it for proteasomal degradation^{6,10,11}. We found co-electroporation of MDM2 can mimic the effect of p53DD in hPSCs by significantly improving the PE or CBE editing efficiency (**new Supplementary Fig. 8h-i**).

We have added these new results in the manuscript on **Page 9 Line 16-25**, and **Page 10 Line 1-7**.

3. The authors have not investigated the role of p53 inhibition on potential off-target effects of these gene editing tools. For example, CBE has been shown to cause deamination of cytosines in DNA and RNA independent of sequence-specified CBE binding (Zuo, et al. Science 2019; Kim, et al. Nature Biotechnology 2017; Grunewald, et al. Nature 2019). How does p53 inhibition impact these off target activities?

Per reviewer's suggestion, we investigated p53DD in CBE induced off-target effects by transcriptome profiling and genome-wide sequencing (WGS). After creating a stop codon in the H2B reading frame by CBE with or without p53DD in the H1-H2B-tdTomato reporter cells, we isolated the edited cell population (tdTomato negative) by flow cytometry (**new Supplementary Fig. 5a**, attached below). RNA-seq analysis of these cells showed similar C-to-U edits with or without p53DD (**new Supplementary Fig. 5b**), indicating that p53DD did not increase off-target effects induced by cytosine deamination of RNA. Similarly, WGS analysis showed that total SNVs as well as C:G to T:A SNVs are comparable with or without p53DD, though under both conditions (with and without p53DD), we observed a slight increase (**new Supplementary Fig. 5c**). These new results are included in the text on **Page 8 Line 1-13**.

Additionally, we assessed genome-wide off-target effect of PE3 editing with or without p53DD. We looked at genome-wide SNVs and indels in two *GBA* mutation corrected, single-cell clones

generated by PE3 (clone #36 and #66), and by PE3 in conjunction with p53DD (clone #1-5 and #27). The parental patient iPSCs (756-iPSCs) were used as the background control for WGS (**new Supplementary Fig. 11a**). We found that total number of new base substitutions or indels were similar between the parental unedited control and PE3 edited or PE3-p53DD edited single-cell clones (**Supplementary Fig. 11b, c**). These data indicate that p53DD improved prime editing efficiency without compromising the genome-wide safety of prime editing. These new results are included in the text on **Page 14 Line 5-14**.

Supplementary Fig. 5

New Supplementary Figure 5. Transcriptome-wide and genome-wide off-target effects were comparable in CBE with or without p53DD.

a. Schematic overview of experimental design used to identify mutations in the H1-H2B-tdTomato cells edited with CBE with or without p53DD by using transcriptome-wide and genome-wide sequencing. Edited cell population (tdTomato negative) was isolated by FACS sorting for RNA-seq and whole genome sequencing (WGS) analysis. Mutations were identified by comparing the sequence of edited cells to that of parental non-edited cells.

b. Number of C-to-U edits induced by CBE (BE3-FNLS) with or without p53DD, as identified in RNA-seq analysis.

c. Total number of single nucleotide variants (SNVs) and number of C:G to T:A SNVs in the CBE edited cells with or without p53DD treatment, as identified in WGS analysis.

New Supplementary Figure 11. p53DD did not increase genome-wide off-target effects in patient's mutation-corrected single-cell iPSC clones.

a. Schematic overview of experimental design to identify genome-wide SNVs and indels in PE3 or PE3 with p53DD edited single-cell iPSC clones using WGS. Point mutation (c. 1226A > G, p. N370S) in the patient-derived iPSC cells was corrected either by PE3 or PE3 with p53DD. Single cell clones were expanded, and the two corrected clones from each condition were validated by Sanger sequencing and subjected to WGS. Parental iPSCs were used as the background control for WGS.

b, c Fold change of SNVs (**b**) and insertions and deletions (indels) (**c**) in PE3 or PE3-p53DD corrected clones as compared to the parental iPSCs. $n=2$ clones with each editing condition.

Other comments:

1. For Miseq data analysis, please specify which mode of CRISPResso2 was used and the specific parameters used.

We now added the mode of CRISPResso2 and the specific parameters in “Method”, as below:

“For all editing quantifications, we set these parameters: Minimum average read quality (phred33 scale) was >30; Exclude 15 bp from the left side and 15 bp from the right side of the amplicon sequence for the quantification of the mutations. The Minimum homology for alignment to an amplicon was set > 60%. sgRNA and pegRNA targets were provided and quantification window was set as -3. To quantify insertion or deletion edits, CRISPResso2 was run in HDR mode using the sequence with desired insertion or deletion editing as the reference sequence. Editing yield was calculated as percentage of perfect HDR aligned reads / total aligned reads. For quantification of point mutation editing, CRISPResso2 was run in standard Cas9 mode. Editing yield was calculated as the percentage of reads containing the edit divided by total reads.”

2. In the sentence from Line 158 to Line 162, the authors concluded that the results suggest low “off-target” potential of PE. Indels at pegRNA nicking sites are not considered as “off-target” activities. I would suggest change “off-target” to “byproduct”.

Thank you for the suggestion. We have changed the “off-target” to “byproduct”, in **Page 7 Line 23**.

Reviewer #3

1. BE can mediate direct conversion of C-to-T or A-to-G and PE can introduce all types of SNP, precise deletion or insertions at the specific site of a genome. In this work, the author described “Transient inhibition of p53 enhances prime editing and base editing efficiencies in human pluripotent stem cells”. But the author didn’t verify the effect of p53DD on PE-mediated mutation/deletion and ABE system. These data should be appended at least in endogenous genome sites, otherwise it was incomplete.

In our original submission, we included data regarding PE-mediated SNP and precise insertion at endogenous loci. These include the disease-relevant SNP induction and correction at *LMNA*, *GBA*, *LRRK2* loci, and 3-bp or 34-bp insertion at *HEK3*. We found that p53DD enhanced the editing efficiency by ~2-fold in hPSCs.

We now added new data to address PE-mediated deletion at endogenous loci. We first evaluated the precise “GT” (2-bp) deletion at *HEK3* locus. Miseq analysis showed that p53DD slightly increased the efficiency of 2-bp deletion for PE2 and significantly increased editing efficiency (> 2-fold) for PE3 (1.64 ± 0.23 % versus 1.18 ± 0.14 % for PE2; 15.94 ± 0.71 % versus 6.95 ± 1.79 % for PE3) (**new Fig. 2a, d**, attached below).

We also evaluated a disease relevant deletion linked to the development of hereditary angioedema in patients, namely the *SERPING1* (c.351delC) mutation²⁰. We use PE2 or PE3 with or without p53DD to introduce *SERPING1* (c.351delC) in H1 hESCs. Miseq analysis showed that p53DD significantly increased the precise Δ“C” in either PE2 or PE3 editing (2.72 ± 0.53 % versus 1.12 ± 0.64 % for PE2; 24.33 ± 1.05 % versus 12.26 ± 0.60 % for PE3), while low frequencies of indel 1 and indel 2 (< 0.2%) were also detected (**new Fig. 2e**, attached below). These results further suggest utility of p53DD in generating disease relevant deletions in hPSCs.

We have added the new results on **Page 10 Line 19-22; Page 11 Line 12-19**.

In our initial testing, we found the ABE editing tools are general inefficient in hPSCs. We believe the general inefficiency of the ABE and limitation of editing window makes it impractical for the purpose of routine generation of disease-relevant isogenic hPSCs. We have now changed “**BE**” into “**CBE**” to be precise throughout the manuscript.

Fig. 2

New Figure 2. p53DD increased PE efficiency for precise insertion and deletion at endogenous loci.

Note: Revised original a, and new added d and e.

a, Schematics of pegRNA design and PE tools for precise "CTT" or "LoxP" insertion or "GT" deletion at HEK3 locus in H1 hESCs. The PegRNA was designed to include a protospacer sequence that targets to the HEK3 locus, a RT template of the 3' extension was designed to include the edit: CTT (3 bp) or LoxP (34 bp) insertion, or "GT" deletion. Additional nicking sgRNA targets were designed for PE3. Listed sequences indicate the "CTT" or "LoxP" insertion or "GT" deletion (on-target products), potential indels at the pegRNA nicking site (indel 1), and potential indels at the nicking sgRNA targeting site (indel 2). On target, Indel 1 and Indel 2 frequencies were analyzed in (b, c, d).

b-d, MISEQ analysis of editing efficiency at the HEK3 locus for "CTT" insertion (b), "LoxP" insertion (c), and "GT" deletion (d) in H1 hESCs using PE tools with/without p53DD, as well as the potential indels frequency. For insertion: n= 3 independent electroporation reactions for PE2 and PE3, without p53DD conditions. n= 2 independent electroporation reactions for PE2 and PE3, with p53DD conditions. For deletion: n= 3 independent electroporation reactions for each condition. Values presented as mean ± S.D. p values were calculated by ordinary one-way ANOVA were ***p < 0.001, ****p < 0.0001.

e. MISEQ analysis of editing efficiency for introducing SERPING1 (c.351delC) in H1 hESCs using PE tools with/without p53DD. n= 2 independent electroporation reactions for PE3 with p53DD conditions. n= 3 independent electroporation reactions for all other conditions. Values presented as mean ± S.D. p values calculated by ordinary one-way ANOVA were ***p < 0.001, ****p < 0.0001.

2. For Supplementary Fig. 5b, c, the author described “the results suggest p53DD increases PE editing efficiencies for “CTT” or “Loxp” insertion at HEK3 locus, without notably increasing indels frequencies”. But there was obvious increase of indels in “CTT” insertion site of PE3. Also, two endogenous sites were too few to conclude the difference of indels accurately. To clearly explain the effect of indel by p53DD, more endogenous genome sites should be tested.

First, we revised the stated sentences to “*We noticed p53DD increase of indels in “CTT” insertion site of PE3 albeit at low levels.*”

Second, we checked four additional endogenous sites for indels after applying our method. We examined indel frequencies at the loci with guide-target mismatch ≤ 3 , compared to the pegRNA or nicking sgRNA targets of PE3 (**new Supplemental Table 5a**, attached below). Miseq analysis showed no or very low levels of indels at those loci in PE3 with or without p53DD (**new Supplemental Table 5b**, attached below). These results suggest co-electroporation with p53DD did not increase indel frequency at those potential off-target loci in PE3 editing for “CTT” insertion. We have added the new results in the text on **Page 11 Line 2-10**.

New Supplemental Table 5- Indel frequency analysis at potential off-target loci in PE3 with or without P53DD editing for “CTT” insertion at HEK3 locus. Related to Fig 2b.

Spacer (5'-3')	PAM	Chromosome	Position	Mismatches(bp)	Off- target
GGCTCAGACTGAGCACCTGA	GAG	chr2	239105064	2	pegRNA-off-1
CACCCAGACTGAGCACGTGC	TGG	chr15	79457589	3	pegRNA-off-2
GAGCCAGAATGAGCACGTGA	GGG	chr10	129794857	3	pegRNA-off-3
GTCCACCAGGAGCCCGGTGC	TAG	chr19	48607506	3	nicking sgRNA-off-1

New supplemental Table 5a: Sequences with guide-target mismatch ≤ 3 , compared to the pegRNA or nicking sgRNA targets. Mismatch targeting sites were found using Benchling: <https://www.benchling.com/>. Red letters indicate the mismatches.

Off-target	Editing Condition	Experiment 1	Experiment 2	Experiment 3	Average
pegRNA-off-1	PE3	0	0	0	0
	PE3+p53DD	0	0	0	0
pegRNA-off-2	PE3	0	0	0	0
	PE3+p53DD	0	0	0	0
pegRNA-off-3	PE3	0.08%	0.06%	0.18%	0.11%
	PE3+p53DD	0.13%	0.14%	0.13%	0.13%
nicking sgRNA-off-1	PE3	0	0	0	0
	PE3+p53DD	0	0	0	0

New supplemental table 5b: The percentage of indel frequency at the cutting site of each off-target loci. Indel frequency were detected using PCR and Miseq. PCR primers were listed in supplemental table 4.

3. In this work, the author isolated and expanded many single-cell clones to detect indels and other bystander products from the PE3+p53DD editing samples. In fact, it may be of no significance. To comprehensively assess the effect of PE3 with or without p53DD, besides Sanger sequencing and Miseq on target sites, genome-wide sequencing analysis of single-cell clone was essential.

Per reviewer’s request, we performed whole genome sequencing (WGS) to identify genome-wide SNVs and indels in two GBA mutation corrected single-cell clones generated by PE3 editing (clone #36 and #66), and in PE3 with P53DD editing (clone #1-5 and #27). The parental patient iPSCs (756-iPSCs) were used as the control (**new Supplemental Fig. 11a**). WGS analysis showed that the total number of new base substitutions or indels was not increased in PE3 edited single-cell clones, with or without p53DD, compared to parental unedited control (**Supplemental Fig. 11b, c**). These data indicate that p53DD improved prime editing efficiency without compromising genome-wide frequency of introducing SNVs or indels. We have added these new results in the manuscript on **Page 14 Line 5-14**.

References

- 1 Hendriks, W. T., Warren, C. R. & Cowan, C. A. Genome Editing in Human Pluripotent Stem Cells: Approaches, Pitfalls, and Solutions. *Cell stem cell* **18**, 53-65, doi:10.1016/j.stem.2015.12.002 (2016).
- 2 Byrne, S. M., Ortiz, L., Mali, P., Aach, J. & Church, G. M. Multi-kilobase homozygous targeted gene replacement in human induced pluripotent stem cells. *Nucleic acids research* **43**, e21, doi:10.1093/nar/gku1246 (2015).
- 3 Giacalone, J. C. *et al.* CRISPR-Cas9-Based Genome Editing of Human Induced Pluripotent Stem Cells. *Curr Protoc Stem Cell Biol* **44**, 5B 7 1-5B 7 22, doi:10.1002/cpsc.46 (2018).
- 4 Bartlett, D. W. & Davis, M. E. Insights into the kinetics of siRNA-mediated gene silencing from live-cell and live-animal bioluminescent imaging. *Nucleic acids research* **34**, 322-333, doi:10.1093/nar/gkj439 (2006).
- 5 Appella, E. & Anderson, C. W. Post-translational modifications and activation of p53 by genotoxic stresses. *Eur J Biochem* **268**, 2764-2772, doi:10.1046/j.1432-1327.2001.02225.x (2001).
- 6 Kubbutat, M. H., Jones, S. N. & Vousden, K. H. Regulation of p53 stability by Mdm2. *Nature* **387**, 299-303, doi:10.1038/387299a0 (1997).
- 7 Komarov, P. G. *et al.* A chemical inhibitor of p53 that protects mice from the side effects of cancer therapy. *Science* **285**, 1733-1737, doi:10.1126/science.285.5434.1733 (1999).
- 8 Zhu, J., Singh, M., Selivanova, G. & Peugot, S. Pifithrin- α alters p53 post-translational modifications pattern and differentially inhibits p53 target genes. *Sci Rep* **10**, 1049, doi:10.1038/s41598-020-58051-1 (2020).
- 9 Zhou, B. P. *et al.* HER-2/neu induces p53 ubiquitination via Akt-mediated MDM2 phosphorylation. *Nat Cell Biol* **3**, 973-982, doi:10.1038/ncb1101-973 (2001).
- 10 Honda, R., Tanaka, H. & Yasuda, H. Oncoprotein MDM2 is a ubiquitin ligase E3 for tumor suppressor p53. *FEBS Lett* **420**, 25-27, doi:10.1016/s0014-5793(97)01480-4 (1997).
- 11 Haupt, Y., Maya, R., Kazaz, A. & Oren, M. Mdm2 promotes the rapid degradation of p53. *Nature* **387**, 296-299, doi:10.1038/387296a0 (1997).
- 12 Yu, J. *et al.* Human induced pluripotent stem cells free of vector and transgene sequences. *Science* **324**, 797-801, doi:10.1126/science.1172482 (2009).
- 13 Toogood, P. L. *et al.* Discovery of a potent and selective inhibitor of cyclin-dependent kinase 4/6. *J Med Chem* **48**, 2388-2406, doi:10.1021/jm049354h (2005).
- 14 Engstrom, J. U. & Kmiec, E. B. Manipulation of cell cycle progression can counteract the apparent loss of correction frequency following oligonucleotide-directed gene repair. *BMC Mol Biol* **8**, 9, doi:10.1186/1471-2199-8-9 (2007).
- 15 Grunewald, J. *et al.* CRISPR DNA base editors with reduced RNA off-target and self-editing activities. *Nature biotechnology* **37**, 1041-1048, doi:10.1038/s41587-019-0236-6 (2019).
- 16 Friedman, P. N., Chen, X., Bargonetti, J. & Prives, C. The p53 protein is an unusually shaped tetramer that binds directly to DNA. *Proceedings of the National Academy of Sciences of the United States of America* **90**, 3319-3323, doi:10.1073/pnas.90.8.3319 (1993).
- 17 Shaulian, E., Haviv, I., Shaul, Y. & Oren, M. Transcriptional repression by the C-terminal domain of p53. *Oncogene* **10**, 671-680 (1995).
- 18 Gencel-Augusto, J. & Lozano, G. p53 tetramerization: at the center of the dominant-negative effect of mutant p53. *Genes Dev* **34**, 1128-1146, doi:10.1101/gad.340976.120 (2020).
- 19 Midgley, C. A. & Lane, D. P. p53 protein stability in tumour cells is not determined by mutation but is dependent on Mdm2 binding. *Oncogene* **15**, 1179-1189, doi:10.1038/sj.onc.1201459 (1997).
- 20 Ferraro, M. F. *et al.* A single nucleotide deletion at the C1 inhibitor gene as the cause of hereditary angioedema: insights from a Brazilian family. *Allergy* **66**, 1384-1390, doi:10.1111/j.1398-9995.2011.02658.x (2011).

REVIEWER COMMENTS

Reviewer #1 (Remarks to the Author):

the authors have addressed all of my concerns, and the manuscript is now significantly clearer and stronger. I recommend publication of this interesting and important work without further delay.

Reviewer #3 (Remarks to the Author):

To authors:

In this revised manuscript, the authors well addressed most of points raised by the reviewers. The manuscript has been substantially improved. Nevertheless, I would like to see the following points to be addressed in the manuscript before its publication in Nature Communication.

Comments/Suggestions,

Reviewer #3 point 2 "For Supplementary Fig. 5b, c, the author described "the results suggest P53DD increases PE editing efficiencies for "CTT" or "Loxp" insertion at HEK3 locus, without notably increasing indels frequencies". But there was obvious increase of indels in "CTT" insertion site of PE3. Also, two endogenous sites were too little to describe the difference of indels accurately. To clearly explain the effect of indel by P53DD, more endogenous genome sites should be selected to analysis it."

In the revised manuscript, the author supplied with the "New Supplemental Table 5" about "Indel frequency analysis at potential off-target loci". In fact, to address this point, the author should provide additional data to evaluate whether the indel frequency is also improved at target sites by using more endogenous genome sites, instead of the indels at potential off-target loci.

Reviewer #4 (Remarks to the Author):

SUMMARY

Li et al. show that CBE and PE efficiencies can be improved (~2-3X) in human pluripotent stem cells by co-delivering a C-terminal, dominant negative fragment of p53 (p53DD). For initial testing of this approach and to show functional BE/PE editing outcomes, the authors sequentially use a vector-based reporter system (gain of GFP signal) and a hESC reporter line with an integrated tomato-based reporter (tomato disruption) to establish that both CBE and PE efficiencies can be improved. They validate their findings using next-generation sequencing (NGS). Furthermore, the authors show that the use of p53DD does not affect CBE-mediated gRNA-independent off-target editing on DNA or RNA. It is also shown that the p53DD effect is not restricted to electroporation-based delivery of plasmids. The authors show that a viral protein co-expressed via the pCE-p53DD vector (EBNA1) does not mediate the abovementioned effect and that co-delivery of MDM2 (a protein involved in p53 degradation) phenocopies p53DD expression, while inhibition of p53 via siRNA or a small molecule inhibitor do not. The PE-p53DD method is used to edit more efficiently at endogenous loci and to generate isogenic hPSC lines. Moreover, the authors perform WGS of PE3-treated cells (with and without p53DD co-expression) which yields no detectable off-target editing. Finally, they show that the p53DD plasmid can accidentally integrate into a cell's genome at a frequency of 2.5% and they provide data that indicates that nicking (and

not Cas9 binding) triggers the p53-p21 pathway and that treatment with drugs that can elicit G1/S cell cycle arrest in hPSCs can decrease CBE/PE efficiency.

ASSESSMENT

P53 inhibition has been used to increase HDR efficiencies in the past, as the authors mention. Nonetheless, I think the work is of considerable importance to anybody working with human pluripotent stem cells and improvements in base and/or prime editing efficiencies are similar to what has been described for MMR inhibition via MLH1dn (Chen et al., Cell 2021). Furthermore, it is interesting to see the beneficial DSB-independent effect of p53-inhibition in the context of CBE and PE use. The authors test their hypotheses carefully, experiments are of high quality, and controls are abundant, especially with respect to editing specificity. I think some minor comments need to be addressed before the study can be published but overall, I think this is great work. Congrats!!

MINOR

1. I wonder if there may be an additive effect of your and Chen et al.'s approach- i.e. transient, combined MMR and p53 inhibition? It would be great to see how PE5 compares to PE3+p53DD in hPSCs/hiPSCs? If the authors have generated these data already, this would make for a nice addition. If not, it might be worth mentioning this as "future steps" in the manuscript.
2. Can the authors explain why they didn't test p53DD co-expression with newer ABEs, like e.g. ABE8e or 8.20? Again, it would be great to see these data, if they were generated. If not, this could be at least commented on in the manuscript. Thanks!
3. Another great control would be translational co-expression of p53DD via 2A. This would also avoid adding another plasmid, thereby minimizing the risk of plasmid integration. Just an idea for future experiments.
4. Enache et al. Nature Genetics 2020 showed that Cas9 editing can lead to the expansion of p53-inactivating mutations. Can the authors elaborate in how far their co-delivery of p53DD is (or isn't) basically a selection of (transiently) p53 deficient cells that have higher "fitness" when being exposed to base/prime editing? I think this is a possibility and it would contradict your sentence on page 16 (lines 12-14) in which you claim that "our data support the action model that co-electroporation of p53DD removes the road block of cell cycle inhibition induced by PE or CBE editing tools, and allows gene editing to proceed more efficiently in hPSCs". How do you prove that gene editing itself is more efficient and that it's not as efficient in cells with active and inactive p53 and that it's just the cell's survival is higher when p53 is inactivated (while being edited)? Just an idea.
5. Do the authors think that this approach might also show benefits when used in primary cells or in vivo?
6. You might consider citing Gehrke et al., Nature Biotechnology 2018 when you first describe eA3A-CBE in the manuscript.

We thank all three reviewers for their careful reading our revised manuscript, their positive comments, and their helpful and constructive suggestions. We have now addressed all their points as detailed below. We hope our responses are satisfactory.

Reviewer #1. the authors have addressed all of my concerns, and the manuscript is now significantly clearer and stronger. I recommend publication of this interesting and important work without further delay.

We greatly appreciate the reviewer's support of the publication of our work!

Reviewer #3. In this revised manuscript, the authors well addressed most of points raised by the reviewers. The manuscript has been substantially improved. Nevertheless, I would like to see the following points to be addressed in the manuscript before its publication in Nature Communication.

Point 2 "For Supplementary Fig. 5b, c, the author described "the results suggest p53DD increases PE editing efficiencies for "CTT" or "Loxp" insertion at HEK3 locus, without notably increasing indels frequencies". But there was obvious increase of indels in "CTT" insertion site of PE3. Also, two endogenous sites were too little to describe the difference of indels accurately. To clearly explain the effect of indel by p53DD, more endogenous genome sites should be selected to analysis it." In the revised manuscript, the author supplied with the "New Supplemental Table 5" about "Indel frequency analysis at potential off-target loci". In fact, to address this point, the author should provide additional data to evaluate whether the indel frequency is also improved at target sites by using more endogenous genome sites, instead of the indels at potential off-target loci.

We appreciate reviewer's clarification of this point. In original manuscript, we have targeted and evaluated 5 different endogenous loci with 9 different mutations using PE with or without p53DD, including the *HEK3* (for insertion and deletion, **Fig.2b-d**), *SERPING1* (for deletion, **Fig 2e**), *GBA* (for SNP induction and correction, **Fig. 3b, e**), *LMNA* (for SNP induction and correction, **Fig 3d, f**) and *LRRK2* (for SNP induction, **Fig.3c**) loci. For each case, we evaluated indel frequencies at the targeting loci, including indel frequency at pegRNA nicking site (indel 1), and nicking sgRNA nicking site (indel 2). Our data showed that p53DD slightly increases indel frequencies, but that those overall frequencies remain low compared to the high precise on-target editing efficiency. In addition, in the 39 disease-relevant isogenic single-cell clones we created using PE3+p53DD, they all carry the desired mutations without carrying any indels or other bystander products around the PE targeting sites, as confirmed by Sanger sequencing (**Supplementary Fig. 10 and 11**).

Per reviewer's suggestion, here we reanalyzed the indel frequencies using *HEK3* locus subjected to "CTT" insertion as an example. Instead of only checking the two endogenous sites, we extended the quantification window to 10bp size which calculated 20 sites adjacent to the pegRNA nicking site (**new Supplemental Fig. 9a**, as shown below). We calculated non-edited frequency, total indel frequency, desired perfect "CTT" insertion, and imperfect "CTT" insertion ("CTT" inserted but carried other mutations) in these 20 sites using CRISPResso2 (**new Supplemental Fig. 9b**, as shown below). Our data showed that PE3 increased both desired and undesired editing frequencies compared to PE2, which is consistent with the previous report¹. The presence of p53DD significantly improved the desired editing efficiency in both PE2 and PE3; the presence of p53DD also increased the total indel and the imperfect "CTT" insertion frequencies, albeit frequencies still represented only a small percentage overall (less than 1% of each). We have now incorporated this data in our manuscript at **Page 11 Line 2-12**.

The newly analyzed results summarized above, in conjunction with indel 1 and indel 2 frequencies in original manuscript, the very low levels of indels at potential off-target loci (Supplemental Table 5), the sanger sequencing of diseases' single-cell clones, as well as the whole genome sequencing analysis, provides comprehensive data for indel frequencies by PE with or without p53DD.

Supplementary Figure 9. Percentage and number of sequence reads were quantified for “CTT” insertion at the HEK3 locus using PE2/3 with or without P53DD. a, Alignment and editing frequencies were calculated in a quantification window (in dotted box) which spanning the -10 position to +10 position relative to the nicking site of pegRNA, Grey bar represents the pegRNA spacer. b, The non-edited frequency and edited frequency including perfect CTT insertion, imperfect CTT insertion as well as other indels within the reading window were calculated using CRISPResso2 with HDR mode. Two independent experiments were conducted, and the data from individual repeat were shown.

Reviewer #4 SUMMARY: Li et al. show that CBE and PE efficiencies can be improved (~2-3X) in human pluripotent stem cells by co-delivering a C-terminal, dominant negative fragment of p53 (p53DD). For initial testing of this approach and to show functional BE/PE editing outcomes, the authors sequentially use a vector-based reporter system (gain of GFP signal) and a hESC reporter line with an integrated tomato-based reporter (tomato disruption) to establish that both CBE and PE efficiencies can be improved. They validate their findings using next-generation sequencing (NGS). Furthermore, the authors show that the use of p53DD does not affect CBE-mediated gRNA-independent off-target editing on DNA or RNA. It is also shown that the p53DD effect is not restricted to electroporation-based delivery of plasmids. The authors show that a viral protein co-expressed via the pCE-p53DD vector (EBNA1) does not mediate the abovementioned effect and that co-delivery of MDM2 (a protein involved in p53 degradation) phenocopies p53DD expression, while inhibition of p53 via siRNA or a small molecule inhibitor do not. The PE-p53DD method is used to edit more efficiently at endogenous loci and to generate isogenic hPSC lines. Moreover, the authors perform WGS of PE3-treated cells (with and without p53DD co-expression) which yields no detectable off-target editing. Finally, they show that the p53DD plasmid can accidentally integrate into a cell's genome at a frequency of 2.5% and they provide data that indicates that nicking (and not Cas9 binding) triggers the p53-p21 pathway and that treatment with drugs that can elicit G1/S cell cycle arrest in hPSCs can decrease CBE/PE efficiency.

ASSESSMENT: P53 inhibition has been used to increase HDR efficiencies in the past, as the authors mention. Nonetheless, I think the work is of considerable importance to anybody working with human pluripotent stem cells and improvements in base and/or prime editing efficiencies are similar to what has been described for MMR inhibition via MLH1dn (Chen et al., Cell 2021). Furthermore, it is interesting to see the beneficial DSB-independent effect of p53-inhibition in the context of CBE and PE use. The authors test their hypotheses carefully, experiments are of high quality, and controls are abundant, especially with respect to editing specificity. I think some minor comments need to be addressed before the study can be published but overall, I think this is great work. Congrats!!

We thank the reviewer for the careful review of our work and the overall positive comments!

MINOR

1. I wonder if there may be an additive effect of your and Chen et al.'s approach- i.e. transient, combined MMR and p53 inhibition? It would be great to see how PE5 compares to PE3+p53DD in hPSCs/hiPSCs? If the authors have generated these data already, this would make for a nice addition. If not, it might be worth mentioning this as "future steps" in the manuscript.

During our revision, a new paper by Chen et al., (mentioned by reviewer #4 above), reported that MMR inhibition via co-delivering MLH1dn improves prime editing efficiencies. We agree that it is very interesting to test if p53DD can enhance the editing efficiency for PE5 or its combination with MLH1dn in hPSCs. We now added this comment as one of the future directions in Discussion at **Page 17, Line 8-10**, as shown below:

"MMR inhibition via co-delivering MLH1dn was recently shown to improve prime editing efficiencies by Chen et al.¹ A future research direction is to examine whether combining this approach with p53 inhibition could further enhance PE efficiency".

2. Can the authors explain why they didn't test p53DD co-expression with newer ABEs, like e.g. ABE8e or 8.20? Again, it would be great to see these data, if they were generated. If not, this could be at least commented on in the manuscript. Thanks!

We found ABE8e was still inefficient in hPSCs in our initial testing. We think the general inefficiency and limitation of editing window of the ABE make it less desirable for routine generation of disease-relevant isogenic hPSCs. In comparison, PE is more versatile and can be applied to precise A-to-G mutations (e.g., for GBA c. 1226A > G in Fig. 3b). We have added the comments about ABE in Discussion on **Page 16, Line 4-7**, as shown below:

“Our initial testing suggested that the ABE editing tools are generally inefficient in hPSCs, making them less desirable for routine generation of disease-relevant isogenic hPSCs. However, it would be interesting to test if p53DD can enhance ABE editing efficiency in other cell types such as primary cells or cancer cells.”

3. Another great control would be translational co-expression of p53DD via 2A. This would also avoid adding another plasmid, thereby minimizing the risk of plasmid integration. Just an idea for future experiments.

This is an excellent idea. We have indeed recently generated the 2A-p53DD vector and preliminary data indicate that it can improve PE editing efficiency. All our vectors will be shared via Addgene.

4. Enache et al. Nature Genetics 2020 showed that Cas9 editing can lead to the expansion of p53-inactivating mutations. Can the authors elaborate in how far their co-delivery of p53DD is (or isn't) basically a selection of (transiently) p53 deficient cells that have higher “fitness” when being exposed to base/prime editing? I think this is a possibility and it would contradict your sentence on page 16 (lines 12-14) in which you claim that “our data support the action model that co-electroporation of p53DD removes the road block of cell cycle inhibition induced by PE or CBE editing tools, and allows gene editing to proceed more efficiently in hPSCs”. How do you prove that gene editing itself is more efficient and that it's not as efficient in cells with active and inactive p53 and that it's just the cell's survival is higher when p53 is inactivated (while being edited)? Just an idea.

Thank you for the reviewer sharing this interesting paper. The paper by Enache et al. studied wild type Cas9 genome editing in human cancer cell lines and their results showed introduction of Cas9 can lead to the emergence and expansion of p53-inactivating mutations. Whether Cas9n-based base editor or prime editor in non-cancer cells, such as hPSC cells can lead to the same effect is currently unclear. On the other hand, p53 transient inhibition was shown to both increase CRISPR-Cas9-based homologous recombination and help to escape the cell cycle arrest caused by gene editing². These findings are in line with our data that p53DD-associated increase in PE/CBE efficiency is linked to blocking the P53-P21 axis that is activated by gene editing tools.

With this said, the current data cannot exclude the possibility that co-delivery of p53DD may increase “fitness: of the overall cell population when exposed editing. We do not think that this would contradict the mechanism that we proposed in the current manuscript, and we have now included this possibility in our proposed model. We revised the sentence (**Page 17 Line 3-7**), as shown below:

“our data support a model whereby co-electroporation of p53DD removes the roadblock of cell cycle inhibition induced by PE or CBE editing tools and increases the editing efficiency of PE or CBE in hPSCs; however, we do not exclude a possible effect of p53DD on increasing overall cell fitness for exposure to these gene editing tools”.

The reviewer raised another interesting question that as to whether p53DD simply functions via improving cell survival. To address this question, further experiment could test and compare the effect of chemicals such as the recently developed cocktail (CEPT)³ which was shown improving hPSCs cell survival in genome editing process, with or without p53DD. Finally, we also cannot rule out the possibility that p53DD directly impacts gene editing itself. For example, an interaction of p53DD with polymerase could be an idea to be examined in future studies, as reviewer 1 previously suggested.

5. Do the authors think that this approach might also show benefits when used in primary cells or in vivo?

We do think that this is a likely possibility, and we have now added this point in the Discussion on **Page 17, Line 10-12**, as shown below:

“It will be also interesting to evaluate whether co-deliver of p53DD facilitates gene editing in primary cells or in vivo.”

6. You might consider citing Gehrke et al., Nature Biotechnology 2018 when you first describe eA3A-CBE in the manuscript.

We have now added this reference on **Page 5 Line 8**. Thank you for the suggestion.

References:

- 1 Anzalone, A. V. *et al.* Search-and-replace genome editing without double-strand breaks or donor DNA. *Nature* **576**, 149-157, doi:10.1038/s41586-019-1711-4 (2019).
- 2 Haapaniemi, E., Botla, S., Persson, J., Schmierer, B. & Taipale, J. CRISPR-Cas9 genome editing induces a p53-mediated DNA damage response. *Nature medicine* **24**, 927-930, doi:10.1038/s41591-018-0049-z (2018).
- 3 Chen, Y. *et al.* A versatile polypharmacology platform promotes cytoprotection and viability of human pluripotent and differentiated cells. *Nature methods* **18**, 528-541, doi:10.1038/s41592-021-01126-2 (2021).

REVIEWERS' COMMENTS

Reviewer #1 (Remarks to the Author):

The authors addressed all my concerns in the previous round. I also looked at the comments of the other reviewers and feel that these have also been addressed carefully. I recommend publication of this important work without further delay.

Reviewer #3 (Remarks to the Author):

In this revised manuscript, the authors have addressed all points raised by the reviewers. The manuscript has been substantially improved, and is suitable for Nature Communications.

Reviewer #4 (Remarks to the Author):

All the points that I had raised have been addressed, thank you!
Therefore, I recommend to publish this manuscript in Nature Communications.
Congrats on this great work.

Julian Grünewald